# Disclosing proteins in the leaves of cork oak plants associated with the immune response to *Phytophthora cinnamomi* inoculation in the roots: A long-term proteomics approach

**Ana Cristina Coelho**[1,2]*, **Rosa Pires**[1], **Gabriela Schütz**[1,3], **Cátia Santa**[4,5], **Bruno Manadas**[4], **Patrícia Pinto**[6]

**1** Center for Electronic, Optoelectronic and Telecommunications (CEOT), University of Algarve, Faro, Portugal, **2** Escola Superior de Educação e Comunicação (ESEC), University of Algarve, Faro, Portugal, **3** Instituto Superior de Engenharia, University of Algarve, Faro, Portugal, **4** CNC—Center for Neuroscience and Cell Biology, University of Coimbra, Coimbra, Portugal, **5** Institute for Interdisciplinary Research (IIIUC), University of Coimbra, Coimbra, Portugal, **6** Center for Marine Sciences (CCMAR), University of Algarve, Faro, Portugal

* acoelho@ualg.pt

**Data Availability Statement:** The mass spectrometry proteomics data have been deposited to the ProteomeXchange Consortium via the

## Abstract

The pathological interaction between oak trees and *Phytophthora cinnamomi* has implications in the cork oak decline observed over the last decades in the Iberian Peninsula. During host colonization, the phytopathogen secretes effector molecules like elicitins to increase disease effectiveness. The objective of this study was to unravel the proteome changes associated with the cork oak immune response triggered by *P. cinnamomi* inoculation in a long-term assay, through SWATH-MS quantitative proteomics performed in the oak leaves. Using the *Arabidopis* proteome database as a reference, 424 proteins were confidently quantified in cork oak leaves, of which 80 proteins showed a p-value below 0.05 or a fold-change greater than 2 or less than 0.5 in their levels between inoculated and control samples being considered as altered. The inoculation of cork oak roots with *P. cinnamomi* increased the levels of proteins associated with protein-DNA complex assembly, lipid oxidation, response to endoplasmic reticulum stress, and pyridine-containing compound metabolic process in the leaves. In opposition, several proteins associated with cellular metabolic compound salvage and monosaccharide catabolic process had significantly decreased abundances. The most significant abundance variations were observed for the Ribulose 1,5-Bisphosphate Carboxylase small subunit (RBCS1A), Heat Shock protein 90–1 (Hsp90-1), Lipoxygenase 2 (LOX2) and Histone superfamily protein H3.3 (A8MRLO/At4G40030) revealing a pertinent role for these proteins in the host-pathogen interaction mechanism. This work represents the first SWATH-MS analysis performed in cork oak plants inoculated with *P. cinnamomi* and highlights host proteins that have a relevant action in the homeostatic states that emerge from the interaction between the oomycete and the host in the long term and in a distal organ.

PRIDE partner repository with the dataset identifier PXD021455.

**Funding:** This work was financially supported by FCT, integrated in projects UID/Multi/00631/2013, UID/Multi/00631/2019 and UIDB/00631/2020 CEOT BASE to CEOT and ACC, GS and RP; UIDB/04326/2020 to CCMAR; fellowship SFRH/BPD/84033/2012 and researcher contract with the University of Algarve under Norma Transitória-DL57/2016/CP1361/CT0015 to PP; contract NIBAP (ALG-01-0247-FEDER-037303) to RP; projects POCI-01-0145-FEDER-007440 (Ref. UIDB/04539/2020), POCI-01-0145-FEDER-016428 (Ref. SAICTPAC/0010/2015), POCI-01-0145-FEDER-029311 (Ref. PTDC/BTM-TEC/29311/2017), POCI-01-0145-FEDER-30943 (Ref. PTDC/MECPSQ/30943/2017) and PTDC/MED-NEU/27946/2017 to CNC, BM and CS. The work at CNC was also funded by the National Mass Spectrometry Network (RNEM) under contract POCI-01-0145-FEDER-402-022125 (Ref. ROTEIRO/0028/2013). The funders had no role in study design, data collection and analysis, decision to publish, or preparation of the manuscript.

**Competing interests:** The authors have declared that no competing interests exist.

# Introduction

The soil-borne oomycete *Phytophthora cinnamomi* infects the roots of cork oak (*Quercus suber*) plants, induces necrotic lesions, and the loss of fine roots [1,2]. This evidence, combined with other factors, are the hallmark for the decline of the cork oak savanna-like ecosystem in Portugal (cork oak *montado*) and Spain (cork oak *dehesa*). Climate changes is reducing water availability (drought) [3], and the effectiveness of roots in absorbing water is affected by the health status of the plant [4,5], which can become less effective in accessing groundwater during drought [6]. Insect colonization [7] and fungal infections [8,9] can weaken the tree's defence system and thus contribute to the decline. To help maintain the sustainability of the cork oak agro-forests, the recommended focus is to adopt good management practices [10].

During inter and intracellular cork oak colonization by *P. cinnamomi*, small 10 kDa proteins (elicitins) are secreted by the oomycete and increases disease effectiveness. This has been demonstrated by studying a β-cinnamomin silenced *P. cinnamomi* strain, which acted as a weaker pathogen against cork oak when compared to the virulence revealed by the wild type [11,12]. In the roots of the narrow-leafed lupin (*Lupinus angustifolius*) infected with *P. cinnamomi*, the expression of *β*-cinnamomin starts to be detected as early as 24 h post-inoculation and follows the development of the mycelium into the host, anchored to a mycelial cell wall protein, emphasizing the recognition of these proteins as virulence factors [13]. However, effector molecules from the RxLR, CRN (for Crinkling and Necrosis) and Nep1-like (NLPs) protein families are also potentially secreted, encoded by the 171 RxLR, 72 NLPs and 29 CRN putative genes present in the genomes (78 Mb) of three *P. cinnamomi* isolates, being able to suppress or bypass the plant basic defence responses [14]. The molecular mechanisms by which the effector molecules act are largely unknown, although the entry of some effector proteins into the plant host cells is known to follow a mechanism of endocytosis after binding to receptor molecules of phosphatidylinositol-3-phosphate (PI-3-P) mediated by the effector RxLR domain [15,16]. In the nucleus, the effectors control reactions that trigger host cell death or hypersensitive responses (HR) [17,18], and in the nucleolus, they can act as modulators of histone acetyltransferases (HAT) to reprogram the plant defence gene expression and promote infection [19].

Following compatible or incompatible reactions with plants, oomycete compounds like lipids or carbohydrates referred to as Pathogen-Associated Molecular Patterns (PAMPs) and effector biomolecules elicit local resistance responses or PAMPs/effector-triggered immunity (PTI/ETI) in their hosts [20]. In *Q. suber* root cells, during the first 24 h of interaction with *P. cinnamomi*, metabolic patterns undergo a non-linear variation for compounds with carbohydrate, glycoconjugate and lipid groups [21]. At the transcriptomic level, the differential expression of genes encoding pathogenesis-related proteins was observed in avocado roots challenged with *P. cinnamomi* [22] and in stem tissues of *Eucalyptus nitens* infected with *P. cinnamomi* [23]. In a more detailed analysis of the transcriptome of chestnut roots inoculated with *P. cinnamomi*, the multiplicity of the defence responses becomes evident with the identification of genes related to the HR (hypersensitive response), cell wall strengthening, synthesis of flavonoids and systemic acquired resistance [24]. Further, resistance (R) genes coding to transmembrane proteins such as LRR receptor-like serine/threonine-protein kinase in two *Castanea* species [24] and CC-NB-LRR (coiled coil-nucleotide binding-leucine rich repeat) in cork oak [25] are also potentially associated to the recognition of effector molecules, eventually interacting, according to the gene-for-gene model [26]. Activation of these resistance proteins can result in the activation of mitogen-activated protein kinase (MAPK) signal transduction cascades, leading to transcription factor activation and transcription of responsive genes, and these cascades can also be activated by proteins sensitive to the production of reactive oxygen species (ROS, $O_2^-$, $H_2O_2$) [reviewed by 20,27–29].

Salicylic acid (SA)/salicylate is also a signaling molecule that plays a central role in PAMPs/effector-triggered immunity (PTI/ETI) and in the systemic acquired resistance (SAR). SAR is a type of immunity that extends to the entire plant beyond the site of infection, protecting the plant against a broad spectrum of pathogens [30,31]. The expression of a large number of pathogenesis-related genes is activated by nuclear transcription factors interacting with NPR1 monomers (nonexpressor of pathogenesis related 1), known as the main regulatory molecule of the SA-signaling pathway [32,33].

To overcome the harmful implications of *P. cinnamomi* on susceptible species of thousands of plants worldwide, one of the current challenges is the identification of molecular markers or physiological processes suitable for recognition of resistant or susceptible host plant species or varieties. Information about the constitutive expression level of pathogenesis-related genes in non-infected hosts and the reaction time mediating the recognition of the invader and the activation of local and systemic defence systems can contribute to this global goal, and was critical for the recognition of *Castanea crenata* [28]. In less susceptible avocado rootstocks, the physical and chemical composition of the host's tissues at the site of infection was critical to the effectiveness of *P. cinnamomi* zoospore germination and penetration, as the early deposition of callose instead of lignin near the site of hyphae penetration along the cell wall hindered the development of the oomycete's hyphae [34].

The hypothesis of the present study is that after inoculation of plant roots with a pathogen, an immune response is initiated that will lead to a new homeostatic state, with protein changes that can be detectable in the long-term, distally from the infection site. The aim was to identify and quantify proteins in the leaves of cork oak plants inoculated with *P. cinnamomi* in the roots and compare them to those in the leaves of non-inoculated plants, at 248 days post-inoculation, using SWATH-MS proteomics [35]. SWATH-MS (Sequential Window Acquisition of all Theoretical Mass Spectra) is a quantitative, label-free and unbiased proteomics method that is able to acquire information about virtually every ion (in this case peptides), introduced into the mass spectrometer [36]. SWATH is a promising strategy for the quantitative screening of a large number of proteins that has previously been applied in the field of plant biology [37–39] and recognized as a valuable tool for the comprehensive study of proteins in plants [40,41].

The leaves are a distal organ that can be sampled in a minimally invasive way in adult trees, so they can also be a potential organ for practical monitoring of infection or resistance. Four hundred and twenty-four proteins were identified in the cork oak leaves, and a subset of 80 proteins showed differential levels between inoculated and control plants, being considered responsive to *P. cinnamomi*. These included 18 proteins associated with several gene ontologies (GO) biological processes, and their potential role in the cork oak immune response is discussed. The GO cellular component "stromules" was also significantly enriched among the differential proteins, indicating that communication between cellular organelles may be important in the cork oak immune response to *P. cinnamomi*.

## Materials and methods

The design of the project included several experimental procedures operated at different time points. In the first phase, the biological material was prepared consisting of twelve cork oak seedlings, germinated from seeds, with half of these plants being inoculated with *P. cinnamomi*. The following phases started 248 days after inoculation and included the harvesting of the leaves from each plant for protein extraction and subsequent SWATH-MS proteomics. The experimental assay ended with the bioinformatic annotation and quantification of proteins present in the extracts of each plant.

## Biological material

Cork oak plants used in this experimental project were germinated from acorns taken from six cork oak trees located in Cachopo, Algarve, Portugal (S1 Fig). Parental cork oak trees referenced as S1.1, S2.1, S4.1, S5.1, S7.1, and S8.1 showed signs of decline at distinct stages of progression, based on visual observation of the canopy defoliation level typical of *P. cinnamomi* infection. The study included two experimental conditions with six biological replicates: 6 cork oak plants inoculated with the PA45 *P. cinnamomi* isolate and 6 non-inoculated plants. Seeds from six parental cork oak trees were germinated and were distributed between the control and inoculated groups so that each inoculated plant had a paired control from the same progenitor. S1 Table provides the cork oak references used in the study. PA 45 was isolated from the rhizosphere of cork oak trees that showed symptoms of decline in the Algarve region and its high virulence on cork oak seedlings was extensively studied [11,12,22]. To reconfirm the identity of the isolate as *P. cinnamomi*, DNA was extracted from PA 45 isolate and was used in PCR reactions with primers (95.422/96.007) designed for a colorimetric molecular assay [42] targeting the elicitin genes (GenBank accession number AJ000071).

For the preparation of control and inoculated plants, twelve 77-day-hold cork oak plants were removed from the germination alveoli, freeing most of the organic substrate that accompanied the roots, and were laid down on trays whose surface was protected with moist absorbent paper. Then, a 2 cm$^2$ agar plug of *P. cinnamomi* mycelium isolate PA45, grown in clarified V8 (Campbell Soup) semi-solid agar, in the dark at 25˚C [11] for 9 days, was placed mycelial surface down on the tap root of 6 cork oak plants—inoculated plants. The roots of the control plants were not exposed to non-colonised semi-solid agar plugs to prevent the growth of microorganisms present in unsterilized roots on the nutritious support (agar surface), whose interaction with plant tissues could elicit defence reactions not present in the natural plants. This situation is prevented in the inoculated samples due to the large amount of *P. cinnamomi* hyphae present on the surface of the agar plugs avoiding bacteria and other microorganisms from having acess to the nutritious support.

The roots of the inoculated and non-inoculated plants were covered with aluminum foil and kept on the moistened trays at 25˚C for 48 hours.

Forty-eight hours after *P. cinnamomi* inoculation the plugs were removed and all plants were potted into a misture of planting soil (PFLANZ-ERDE) and sand (proportion 2/3 for 1/3) in free-drining plastic containers (Top Ø 16 cm; Base Ø 13 cm; H 33 cm), transferred outside and watered regularly to container capacity. S1 Fig outlines the procedure and timing of the experiment. After 248 days, the leaves of cork oak plants, inoculated and non-inoculated, were collected and immediately frozen in liquid nitrogen and stored at -80˚C until further use for protein extraction.

## SWATH-MS proteomics

**Total protein extraction.** The optimized extraction of proteins from cork oak leaves included eight steps. 1) Leaf tissue (200 mg) was ground in a mortar and pestle in the presence of liquid nitrogen to obtain a fine powder. 2) Buffer 1 (1.25 mL/100 mg sample; DTT–Dithiothreitol 60 mM; 10% TCA-Trichloroacetic acid solubilized in acetone) was added to the mortar and samples were macerated in the presence of the buffer with the pestle. 3) The heterogeneous solution was transferred to a 2 mL microcentrifuge tube and incubated for 1 hour at -80˚C. 4) The samples were centrifuged at 15,000× g for 15 min at 4˚C and the supernatant was discarded. 5) The pellets were dissolved in 1 mL of Buffer 2 (2.5 ml/100 mg sample; DTT 60 mM solubilized in acetone), and the sample volume was divided into two microcentrifuge tubes followed by the addition of 750 μL of Buffer 2 to each tube. 6) These solutions were

incubated for 1 hour at -80˚C. Procedures 5 and 6 were repeated until the solution was clear green. 7) The samples were centrifuged at 15,000×g for 15 min at 4˚C and the supernatant was discarded. 8) Pellets were dried and resuspended in 250 μL of SDS-PAGE buffer [TRIS Glycine buffer solution (25 mM TRIS; 192 mM Glycine; Sigma-Aldrich); 2% SDS-Sodium dodecyl sulfate] followed by incubation at 95˚C for 5 min and centrifugation at 20,000x g for 15 min at 4˚C. All reagents used were molecular biology grade.

Protein concentration in the samples was estimated using the 2D-Quant kit (GE Healthcare Life Sciences) with serum albumin as standard [43].

**SWATH-MS strategy.** For the proteomic screening, the short GeLC-SWATH-MS strategy was used according to [44] with minor modifications. Briefly, 50 μg of each sample and a pooled sample per group (pool of the protein extracts for the six control or six inoculated samples) were subjected to *in-gel* digestion after a partial SDS-PAGE run. Then, LC-MS information was acquired in two different acquisition modes: information-dependent acquisition (IDA) of the pooled samples, and SWATH-MS (Sequential Windowed data-independent Acquisition of the Total High-resolution Mass Spectra) of each sample. Protein identification and library construction was performed using ProteinPilot™ (v5.0.1, Sciex), and compared with the *Arabidopsis thaliana* reference proteome (retrieved from https://www.uniprot.org/ in April 2018). In addition, protein identification was tested against the predicted proteins deduced from the recently published draft genome sequence of cork oak [45], available at http://corkoakdb.org/downloads (fileGCF_002906115.1_CorkOak1.0_protein.faa, accessed in November 2020). The relative quantification was performed using the SWATH™ processing plug-in for PeakView™ (v2.2, Sciex). For each experimental group, the average protein levels, standard deviation and percentage coefficient of variation (% CV) were calculated based on the quantification levels obtained for each individual with six biological replicates per group. The fold change (FC) between inoculated and control plants was calculated by dividing the respective median protein levels for all quantified proteins. Statistical comparisons between protein levels were carried out using the software SPSS v23 (IBM) and the non-parametric Mann Whitney U-test (MW). Proteins were considered as differentially modified when FC was greater than 2 or less than 0.5 or MW p-value was below 0.05.

The mass spectrometry proteomics data have been deposited to the ProteomeXchange Consortium via the PRIDE [42] partner repository with the dataset identifier PXD021455. A detailed description of LC-MS materials and methods are provided as supporting information (S1 File).

**Enrichment analyses and hierarchical clustering.** Gene ontology (GO) and pathway (KEGG and Reactome) enrichment analyses were carried out as in [46,47], using Cytoscape v3.5.1 and ClueGO plug-in v2.5.2 [48,49], comparing the list of 80 differentially modified proteins against the *Arabidopsis thaliana* [organism 3702] set of GO biological process and cellular component databases from November 2017. Enrichment analyses were repeated using the databases updated in 2020 and the same general enriched terms were found (data not shown). The following settings were used for the ClueGO enrichment analysis (right-side): GO levels 3 to 8, Benjamini-Hochberg false discovery rate (FDR) correction with a cut-off at FDR<0.05 and minimum of three genes/4% for terms to be considered significant. The initial group size was set as 1, group merging at 50%, and Kappa-statistics score at 0.4.

Enrichment scores of the functionally related network groups were calculated as -Log2 [group FDR]. The leading terms of each enriched group were those with the lowest term FDR (highest enrichment score), which was used to name the respective group.

The hierarchical clustering of the 80 differentially modified proteins was analysed with Cluster 3.0 at http://bonsai.hgc.jp/~mdehoon/software/cluster/software.htm#ctv using normalized protein levels, applying the uncentered correlation and complete linkage options.

## Results and discussion

### Observation of the plants

Cork oak plants were in contact with the *P. cinnamomi* mycelium at the beginning of the experiment for 48 hours, with no (re)inoculation over the next eight months until the end of the experimental assay. During the first 24 hours of inoculation with *P. cinnamomi* strain PA45, the aerial apex of the inoculated plants wilted and after 48 hours, all the inoculated roots appeared necrotic at the inoculation site (S2 Fig). At this time there was no observable changes in the control plants. Seven months after inoculation, 1 month before the end of the experiment, it was not possible to distinguish control plants from plants inoculated with *P. cinnamomi* by visual observation of the aerial part (S2 Fig). The vegetative development of the plants looked similar in both experimental conditions, inoculated and non-inoculated. Although no foliar symptoms of *P. cinnamomi* infection were observed, the infection is expected to have spread beyond the inoculation site through zoospores released from sporangia who migrated into the irrigation water or through root to root contact.

The virulence of the PA45 strain had been previously tested in cork oak roots, inoculated under the same conditions as in the present study for 3 days [11]. Histological studies performed on colonized root tissue demonstrated the ability of the oomycete to invade the epidermis, cortical parenchyma and vascular cylinder both inter- and intra-cellularly, and to destroy host cells [11].

In nature, at infested sites, cork oak trees may succumb (sudden death) after the summer, without showing obvious previous symptoms of decline, or they can remain for years with symptoms of defoliation that slowly worsen over time (slow decline). *P. cinnamomi* has been isolated from roots of declined symptomatic trees and from infested soils throughout Portugal, and it is important to recognize that oomycete infection can be a determining factor for cork oak decline. However, the recovery of *P. cinnamomi* from declining trees does not provide information about the plants' responsiveness or vitality over time.

At the end of this experiment, the plants inoculated with *P. cinnamomi* in the form of a single event were visually asymptomatic for leaf fall or yellowing, plant height or number of leaves. Nevertheless, the molecular interaction between *P. cinnamomi* and the hosts may have occurred differently in each of the six plants due to the high molecular diversity characteristic of *Q. suber* species [50]. One of the pertinent questions is how to detect that a plant is or has been invaded/infected when it has no symptoms, avoiding the (re)isolation of the pathogen and the use of invasive methods. As the degree of tree defoliation is a symptom of decline and leaf harvesting is a method minimally invasive to adult trees, the search for molecular markers in the leaves of plants challenged with *P. cinnamomi* can be a valuable option. The defence responses induced in the host by *P. cinnamomi* in the long term and distant from the inoculation site establish a homeostatic state adapted to living with the invader. This new homeostatic state stands out when comparing (below) the type and amount of proteins present in the leaves of inoculated and non-inoculated plants.

### Leaf cork oak proteome changes in response to *P. cinnamomi* inoculation

SWATH-MS Proteomics analysis performed on leaf samples was used to characterize the proteome of cork oak plants inoculated with *P. cinnamomi*, and compared with the proteome of non-inoculated plants. With this technique, 12 individual protein profiles were obtained, and protein abundances were quantified in each of the leaf extracts. Thus, the protein profiles obtained for the six biological replicates in the two experimental conditions (control and inoculated) reflect the genetic variability of the *Q. suber* species, assuming the average of the results a value closer to reality.

Four proteome databases were used for a comprehensive protein sequence catalogue and to compare their differential abundance. Table 1 shows the number of identified or quantified

proteins in the samples with reference to the Plant proteome (containing all plant entries in the SwissProt database), or reference proteomes for *Populus trichocarpa* and *Arabidopsis thaliana* contained in the Uniprot database, as well as to the proteins deduced from the first draft genome of *Quercus suber* from CorkOakDB (release 2018) [45].

The identification and quantification proteomics results are presented for the analyses using the Uniprot reference proteome for *Arabidopsis* (S2 Table) and for the predicted proteins from the cork oak genome (S3 Table). The later provided a probable annotation to 1,388 predicted proteins obtained by information-dependent acquisition (IDA) from pooled samples for each group. However, 58.8% of these proteins matched protein predictions of low confidence (containing the designations -like, -probable, -uncharacterized or -low quality protein) or corresponded to repeated entries among the CorkOakDB predicted proteins, revealing a high redundancy in this database. Consequently, the quantification of 841 predicted oak proteins was of low confidence, as the shared peptides were not able to be quantified under the quality criteria used for SWATH (S3 Table).

Given the robustness of the *Arabidopsis* protein database (reference proteome available at Uniprot.org, an highly curated protein database with low frequency of proteins of unknown function) and the availability of substantial functional annotation for gene ontologies and pathways, the *Arabidopsis thaliana* (considered a model organism for plants) was chosen as a reference for the following analyses. The exercise of inferring a biological meaning for proteins that stand out in the context of the interaction between *Q. suber* and *P. cinnamomi* it is only achievable taking as a reference a database with evidence-based functional annotation.

Thus, using the *Arabidopis* proteome database as reference, 424 proteins were confidently quantified in the cork oak leaves, with six biological replicates for each of the conditions control or inoculated (S2 Table). From these, 80 proteins showed a fold-change greater than 2 (or less than 0.5, in the case of proteins with decreased levels) or a p-value below 0.05 in their median levels between inoculated and control samples (Table 2). The Venn diagram in Fig 1 shows the number of proteins that met one or both criteria.

Among the 80 proteins with differential levels, 60 proteins increased abundance, and 20 proteins decreased abundance in the leaves of inoculated cork oak plants, 8 months after *P. cinnamomi* inoculation, compared to the control plants (Table 2).

## Hierarchical clustering of differentially produced cork oak proteins

The proteins with differential levels between inoculated and control samples were clustered in a heatmap to allow better visualization of the protein variation patterns (Fig 2). Inoculated plants are clearly distinguished from control plants based on the profiles of this protein dataset. In other words, eight months after a single inoculation of the cork oak root with *P. cinnamomi*,

**Table 1. Number of proteins identified and quantified.**

| Number of identified proteins (5% local-FDR)[a] | Reference proteome database | Number of quantified proteins (5% local-FDR) |
|---|---|---|
| 802 | Plant (SwissProt database) | 523 |
| 783 | *Populus trichocarpa* (UP000006729) | 536 |
| 608 | *Arabidopsis thaliana* (UP000006548) | 424 |
| 1,388 | *Quercus suber* (CorkOakDB) | 841 |

[a]A local false discovery rate of 5% was used as criteria for acceptance of peptide assignments and protein identifications.

**Table 2.  List of differentially accumulated proteins in *Q. suber* leaf proteome 8 months after *P. cinnamomi* inoculation, using the *Arabidopis* proteome database as a reference.**

| *Arabidopis* UniProt accession[a] | Median C (x10$^{-3}$)[b] | Median I (x10$^{-3}$)[b] | p≤0.05[d] | log$_2$FC[c] | Protein name[a] | Protein Initials[a] |
|---|---|---|---|---|---|---|
| Proteins more abundant in *P. cinnamomi* inoculated samples compared to the control | | | | | | |
| P27323 | 0.056 | 0.169 | 0.015 | 1.7 | Heat shock protein 90–1 | HS90-1 |
| Q9FIF3 | 0.120 | 0.274 | 0.041 | 1.4 | 40S ribosomal protein S8-2 | RS82 |
| O81644 | 0.020 | 0.037 | 0.132 | 1.3 | Villin-2 | VILI2 |
| P38418 | 0.188 | 0.529 | 0.24 | 1.2 | Lipoxygenase 2, chloroplastic | LOX2 |
| A0A1P8AWT7 | 0.533 | 1.361 | 0.041 | 1.1 | Catalase 3 | A0A1P8AWT7 |
| Q940B0 | 0.232 | 0.436 | 0.065 | 1.1 | 60S ribosomal protein L18-3 | RL183 |
| A8MRL0 | 0.183 | 0.397 | 0.015 | 1.1 | Histone superfamily protein | A8MRL0 At4G40030 |
| Q9FGX1 | 0.124 | 0.260 | 0.004 | 0.9 | ATP-citrate synthase beta chain protein 2 | ACLB2 |
| O04499 | 0.098 | 0.195 | 0.009 | 0.9 | 2,3-bisphosphoglycerate-independent phosphoglycerate mutase 1 | PMG1/iPGAM |
| O49485 | 0.575 | 1.064 | 0.041 | 0.9 | D-3-phosphoglycerate dehydrogenase 1, chloroplastic | SERA1 |
| Q9LF37 | 0.038 | 0.074 | 0.041 | 0.9 | Chaperone protein ClpB3, chloroplastic | CLPB3 |
| Q9STX5 | 0.175 | 0.389 | 0.041 | 0.9 | Endoplasmin homolog | ENPL |
| Q9M040 | 0.171 | 0.336 | 0.009 | 0.8 | Pyruvate decarboxylase 4 | PDC4 |
| Q9SIH0 | 0.142 | 0.257 | 0.004 | 0.8 | 40S ribosomal protein S14-1 | RS141 |
| Q9SIM4 | 0.302 | 0.495 | 0.015 | 0.7 | 60S ribosomal protein L14-1 | RL141 |
| Q93ZN2 | 0.282 | 0.437 | 0.041 | 0.7 | Probable aldo-keto reductase 4 | ALKR4 |
| Q9LKR3 | 1.119 | 1.835 | 0.041 | 0.7 | Mediator of RNA polymerase II transcription subunit 37a | MD37A |
| Q9FMP3 | 1.183 | 2.187 | 0.026 | 0.7 | Dihydropyrimidinase | DPYS |
| Q9S9N1 | 1.651 | 2.616 | 0.004 | 0.7 | Heat shock 70 kDa protein 5 | HSP7E/BiP1 |
| P42798 | 0.320 | 0.489 | 0.004 | 0.6 | 40S ribosomal protein S15a-1 | R15A1 |
| A8MS03 | 0.126 | 0.199 | 0.026 | 0.6 | Ribosomal protein S6 | A8MS03 |
| A8MS28 | 0.481 | 0.754 | 0.026 | 0.6 | Ribosomal L27e protein family | A8MS28 |
| Q9SEI3 | 0.326 | 0.496 | 0.026 | 0.6 | 26S proteasome regulatory subunit 10B homolog A | PS10A/RTP4A |
| Q9SII0 | 0.257 | 0.395 | 0.009 | 0.6 | Probable histone H2A variant 2 | H2AV2 |
| Q39142 | 2.316 | 3.472 | 0.041 | 0.6 | Chlorophyll a-b binding protein, chloroplastic | Q39142 |
| P16181 | 0.224 | 0.332 | 0.009 | 0.5 | 40S ribosomal protein S11-1 | RS111 |
| Q9SRV5 | 2.348 | 3.690 | 0.041 | 0.5 | 5-methyltetrahydropteroyltriglutamate-homocysteine methyltransferase 2 | METE2 |
| P49107 | 0.604 | 1.064 | 0.004 | 0.5 | Photosystem I reaction center subunit N, chloroplastic | PSAN |
| P59259 | 9.892 | 14.422 | 0.041 | 0.5 | Histone H4 | H4/HIS4 |
| Q9LHA8 | 0.325 | 0.467 | 0.015 | 0.5 | Probable mediator of RNA polymerase II transcription subunit 37c | MD37C |
| O04486 | 0.251 | 0.354 | 0.041 | 0.5 | Ras-related protein RABA2a | RAA2A |
| P59233 | 3.624 | 5.261 | 0.015 | 0.5 | Ubiquitin-40S ribosomal protein S27a-3 | R27AC |
| Q8W4H7 | 7.269 | 10.076 | 0.015 | 0.5 | Elongation factor 1-alpha 2 | EF1A2 |
| P52577 | 2.145 | 3.732 | 0.002 | 0.5 | Isoflavone reductase homolog P3 | IFRH |
| F4JWF7 | 0.616 | 0.849 | 0.041 | 0.5 | DEAD/DEAH box RNA helicase family protein | F4JWF7 |
| Q9SVR0 | 0.114 | 0.183 | 0.009 | 0.5 | 60S ribosomal protein L13a-3 | R13A3 |
| P59224 | 2.369 | 3.154 | 0.009 | 0.5 | 40S ribosomal protein S13-2 | RS132 |
| Q9SRZ6 | 0.376 | 0.563 | 0.026 | 0.4 | Cytosolic isocitrate dehydrogenase [NADP] | ICDHC/cICDH |
| Q9LZH9 | 0.282 | 0.384 | 0.009 | 0.4 | 60S ribosomal protein L7a-2 | RL7A2 |
| Q9LD28 | 0.620 | 1.017 | 0.041 | 0.4 | Histone H2A.6 | H2A6 |
| Q948K6 | 0.232 | 0.315 | 0.015 | 0.4 | Ras-related protein RABG1 | RABG1 |
| P22953 | 0.397 | 0.573 | 0.026 | 0.4 | Probable mediator of RNA polymerase II transcription subunit 37e | MD37E |

*(Continued)*

**Table 2.** (Continued)

| *Arabidopsis* UniProt accession[a] | Median C (x10⁻³)[b] | Median I (x10⁻³)[b] | p≤0.05[d] | log₂FC[c] | Protein name[a] | Protein Initials[a] |
|---|---|---|---|---|---|---|
| Q9SU58 | 0.146 | 0.209 | 0.015 | 0.4 | ATPase 4, plasma membrane-type | PMA4 |
| Q8H156 | 1.697 | 2.498 | 0.009 | 0.4 | GTP-binding nuclear protein Ran-3 | RAN3 |
| A0A1P8B2Y6 | 0.197 | 0.287 | 0.026 | 0.4 | Plasma membrane ATPase | A0A1P8B2Y6 |
| A8MS75 | 1.813 | 2.570 | 0.026 | 0.4 | Chlorophyll a-b binding protein, chloroplastic | A8MS75 |
| Q9FJA6 | 1.027 | 1.379 | 0.004 | 0.4 | 40S ribosomal protein S3-3 | RS33 |
| Q9FF90 | 0.841 | 1.131 | 0.015 | 0.4 | 60S ribosomal protein L13-3 | RL133 |
| F4J3P1 | 1.255 | 1.589 | 0.004 | 0.4 | Ribosomal protein L14p/L23e family protein | F4J3P1 |
| Q9LXG1 | 0.243 | 0.382 | 0.026 | 0.4 | 40S ribosomal protein S9-1 | RS91 |
| Q6ICZ8 | 0.167 | 0.250 | 0.002 | 0.3 | Nascent polypeptide-associated complex subunit alpha-like protein 3 | NACA3 |
| P0CJ47 | 0.700 | 0.970 | 0.026 | 0.3 | Actin-3 | ACT3 |
| Q9SZ54 | 0.346 | 0.476 | 0.041 | 0.3 | Putative glutathione peroxidase 7, chloroplastic | GPX7 |
| A0A1P8B767 | 0.610 | 0.773 | 0.041 | 0.3 | Quinone reductase family protein | A0A1P8B767 |
| Q8LB10 | 0.117 | 0.141 | 0.015 | 0.3 | ATP-dependent Clp protease proteolytic subunit-related protein 4, chloroplastic | CLPR4 |
| F4JJ94 | 0.801 | 1.008 | 0.026 | 0.3 | General regulatory factor 1 | F4JJ94 |
| Q9LUD4 | 0.326 | 0.414 | 0.041 | 0.3 | 60S ribosomal protein L18a-3 | R18A3 |
| Q93VH9 | 2.069 | 2.492 | 0.015 | 0.2 | 40S ribosomal protein S4-1 | RS41 |
| O23254 | 0.837 | 1.142 | 0.041 | 0.2 | Serine hydroxymethyltransferase 4 | GLYC4 |
| O49299 | 2.435 | 3.204 | 0.002 | 0.2 | Probable phosphoglucomutase, cytoplasmic 1 | PGMC1/PGM1 |
| *Proteins less abundant in P. cinnamomi inoculated samples compared to the control* | | | | | | |
| F4J3Q8 | 0.345 | 0.099 | 0.004 | -3.5 | P-loop containing nucleoside triphosphate hydrolases superfamily | F4J3Q8 |
| P10795 | 0.878 | 1.543 | 0.818 | -3.1 | Ribulose bisphosphate carboxylase small chain 1A, chloroplastic | RBS1A/RBCS1A |
| Q9FLN4 | 0.218 | 0.100 | 0.132 | -1.5 | 50S ribosomal protein L27, chloroplastic | RK27 |
| Q9FZ47 | 1.255 | 0.641 | 0.065 | -1.3 | ACT domain-containing protein ACR11 | ACR11 |
| O04603 | 0.412 | 0.201 | 0.041 | -1.2 | 50S ribosomal protein L5, chloroplastic | RK5 |
| Q8RX32 | 0.513 | 0.260 | 0.026 | -0.6 | Tropinone reductase homolog At1g07450 | TRNH2 |
| Q9SCW1 | 0.184 | 0.110 | 0.026 | -0.6 | Beta-galactosidase 1 | BGAL1 |
| F4JYM8 | 0.543 | 0.294 | 0.026 | -0.6 | Thiolase family protein | F4JYM8 |
| A0A1P8B485 | 0.402 | 0.257 | 0.015 | -0.6 | Protein translocase subunit SecA | A0A1P8B485 |
| P25697 | 8.890 | 6.489 | 0.015 | -0.6 | Phosphoribulokinase, chloroplastic | KPPR/PRK |
| Q9LRR9 | 1.487 | 0.662 | 0.002 | -0.6 | (S)-2-hydroxy-acid oxidase GLO1 | GLO1/GOX1 |
| B3H4S6 | 0.434 | 0.300 | 0.041 | -0.3 | Dicarboxylate transporter 1 | B3H4S6 |
| P56778 | 19.522[b] | 15.325[b] | 0.002 | -0.3 | Photosystem II CP43 reaction center protein | PSBC |
| P56761 | 14.189 | 11.445 | 0.015 | -0.3 | Photosystem II D2 | PSBD |
| Q9LF98 | 2.338 | 1.901 | 0.041 | -0.3 | Fructose-bisphosphate aldolase 8, cytosolic | ALFC8/FBA8 |
| F4KDZ4 | 2.902 | 1.688 | 0.026 | -0.3 | Malate dehydrogenase | F4KDZ4/PMDH2 |
| Q42525 | 0.553 | 0.340 | 0.026 | -0.3 | Hexokinase-1 | HXK1 |
| P27140 | 7.947 | 6.602 | 0.041 | -0.2 | Beta carbonic anhydrase 1, chloroplastic | BCA1 |
| A0A1P8BG37 | 3.388 | 2.750 | 0.041 | -0.2 | Photosystem II stability/assembly factor, chloroplast | A0A1P8BG37 |
| Q9SAU2 | 1.863 | 1.309 | 0.026 | -0.2 | D-ribulose-5-phosphate-3-epimerase | Q9SAU2/RPE |

[a]UniProt accession, protein name and protein initials arise from the annotation using the *Arabidopsis* proteome database as a reference. For more details on the abundance levels per replicate consult S2 Table.

[b]Data generated from SWATH-MS proteomics: median peak areas for 6 control cork oak plants (Median C) and 6 plants inoculated with *P. cinnamomi* (Median I).

[c]Fold change ratio logarithm of protein abundance in inoculated over control samples greater than 1 or less than -1 (Log2FC).

[d]Non-parametric Mann Whitney U-test (MW) with statistical significance level set to less than 5% (p<0.05).

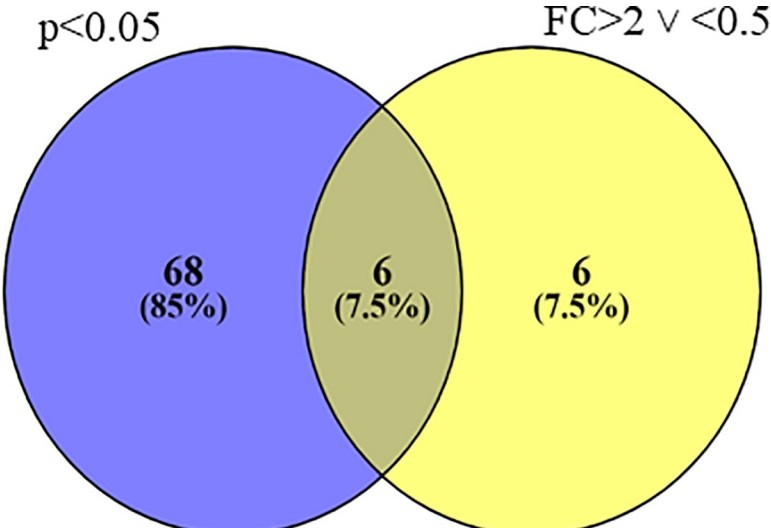

**Fig 1. Protein groups.** The Venn diagram illustrates the number of proteins with a fold-change greater than 2 or less than 0.5 (yellow colour), those with a p-value below 0.05 (blue colour), and those that meet simultaneously both criteria based on protein levels between inoculated and control samples.

the inoculated but asymptomatic plants revealed a leaf proteome significantly different from the non-inoculated plants.

Two scenarios are possible for the inoculated plants: 1) the development of the oomycete was restricted to the inoculation site, with no spread of the infection; or 2) the development of the oomycete took place beyond the inoculation site, invading other tissues, but the infection still did not affect the vegetative state of the host. For the first hypothesis, the protein profiles observed in the leaves may be the result of the activation of the systemic defence system, maintained in memory over time. But, for the second hypothesis, the protein profiles of the inoculated plants may denote a homeostatic state of continuous interaction with *P. cinnamomi*.

The evaluation of cork oak infection by *P. cinnamomi* are always assessed at the root level, in a qualitative way, requiring an experienced technician for the identification of necrosis and/ or absence of feeder roots. But, in this experiment, attempts were made to mimic field conditions, which are hampered by limitations regarding the detection and quantification of the oomycete in the rhizosphere of the trees. All the inoculated plants were used at the end of the experiment, and there was no selection based on the re-isolation of the oomycete or the existence of infection symptoms like leaf yellowing and wilting. Assessing cork oak decline in the field is based on the degree of canopy defoliation, and even if *P. cinnamomi* is isolated from the roots of declining trees, it is not possible to know the level or time of infection. Furthermore, the current methods used to isolate and identify *P. cinnamomi* from the rizosphere of oak roots are based on baiting tecnhiques, pathogen growth in selective media and molecular identification with specific primers. These procedures are time consuming, require expertise and are of relatively low effectiveness. Thus, evaluating cork oak decline through the leaf immune response protein profile induced by *P. cinnamomi* inoculation establishes a new approach for understanding the importance of this oomycete to cork oak decline.

## Association of proteins to GO functional categories and biological pathways

Enrichment analysis was carried out on the selected dataset of 80 differential proteins, which assigned several GO terms to the proteins (based on the *Arabidopsis thaliana* proteins

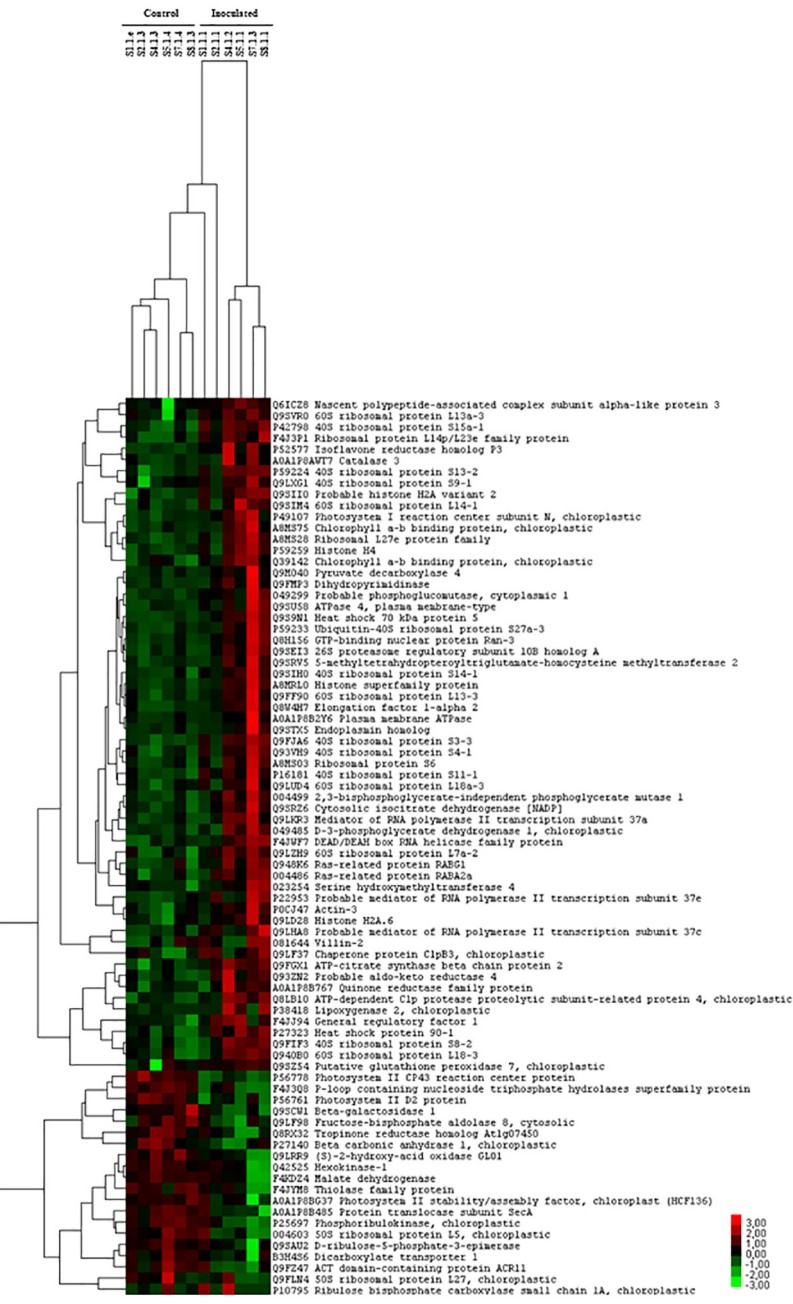

**Fig 2. Hierarchical clustering of differentially produced cork oak proteins.** The heat map clusters the expression patterns of the 80 proteins with altered abundances between inoculated and control plants. Each column represents one cork oak plant; the control (C) plants are the first 6 columns on the left, and the 6 columns on the right are the inoculated (I) plants. The first 60 lines starting from the top of the heat map are proteins with an increased level in the inoculated samples (red color code) and the 20 lines towards the bottom, are proteins with a decreased level in the inoculated samples (green color code). The color scale of the heat map ranges from -3 to 3 (from light green to red).

functional annotations), integrating them into the Gene Ontology functional categories of "Biological Process" (GO_BP) or "Cellular Component" (GO_CC). Within each classification, the significantly enriched terms (FDR<0.05; see lists in S4 and S5 Tables) were assembled into groups of functionally related terms by Cytoscape/ClueGO analysis and the most significantly enriched groups are presented in Fig 3 and summarized in S6 and S7 Tables.

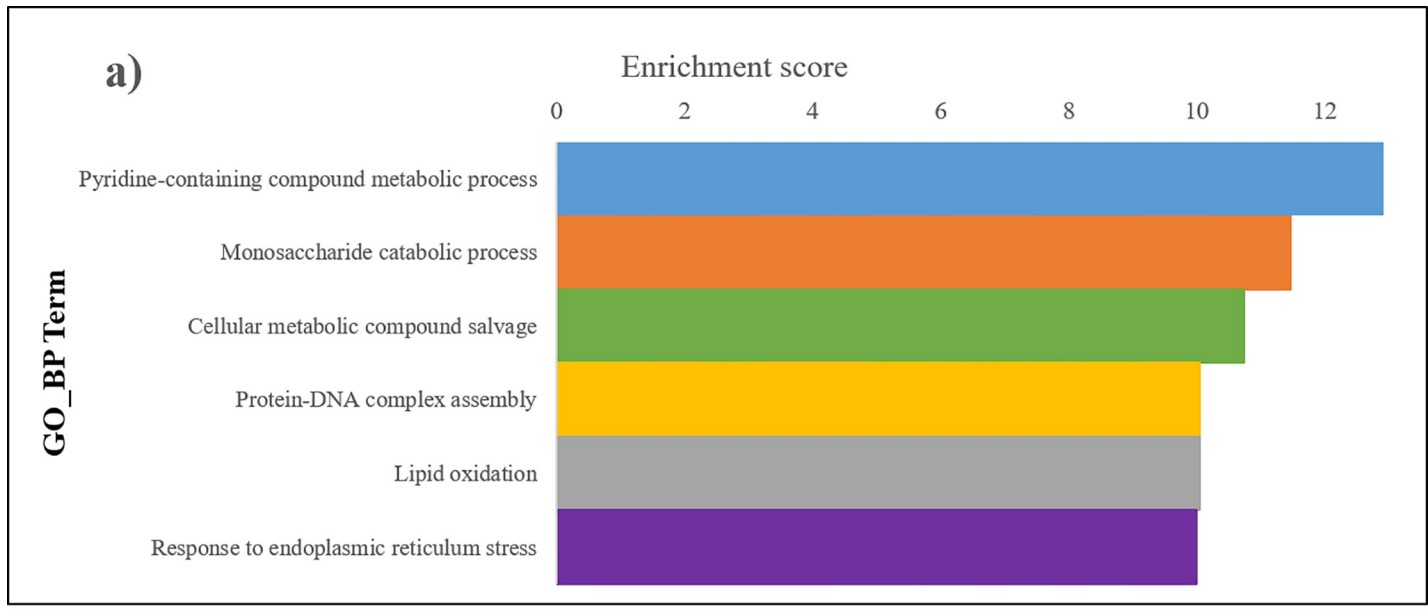

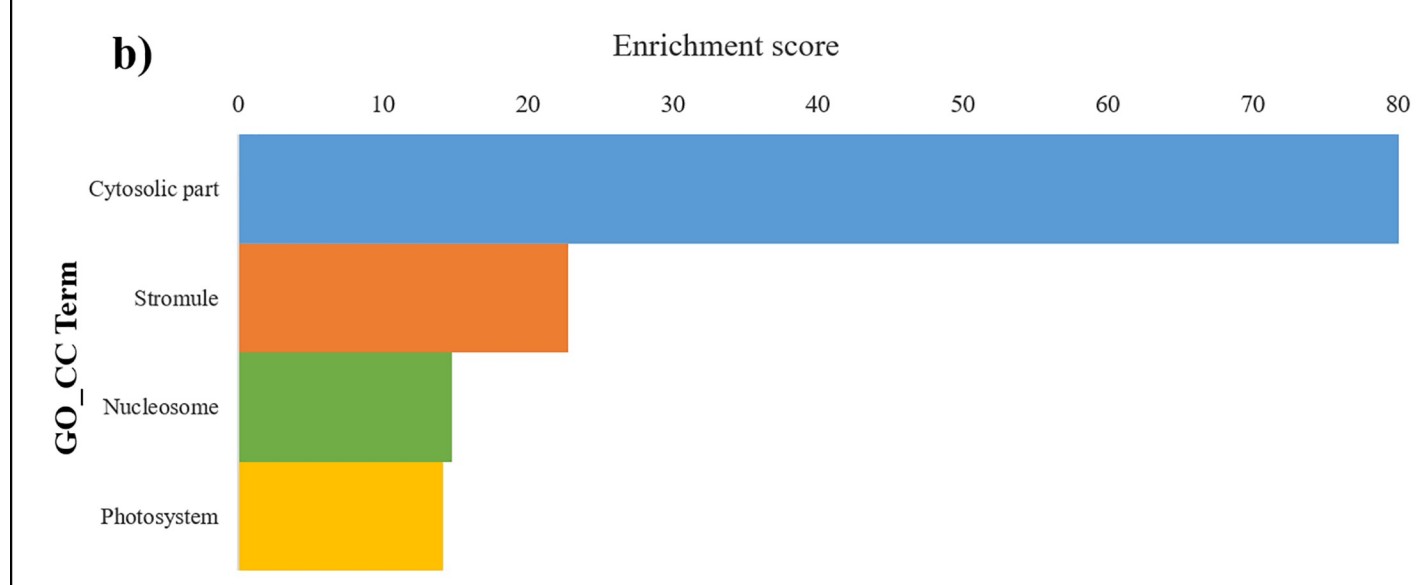

**Fig 3. Enrichment analysis applied to the subset of 80 differential proteins.** The bars represents the groups with higher enrichment score [−log2 (group FDR)] obtained for each group of functional related GO terms; the enrichment of the Gene Ontology category of GO_BP are showed in panel a), and those for GO_CC in panel b). Each group is labelled by the most significant (<FDR) enriched term, used as representative of the total enriched terms in each group that can be consulted in detail in S4 and S5 Tables.

Thirty six GO Biological Process terms were significantly enriched among the differential proteins (S4 Table), which were grouped into 6 groups of functional related GO terms (summarized in Fig 3A); the group *Pyridine-containing compound metabolic process* has the highest enrichment score of all groups (12.9, corresponding to an FDR of $1.29 \times 10^{-4}$), followed by *Monosaccharide catabolic process* (11.5), *Cellular metabolic compound salvage* (10.7), *Protein-DNA complex assembly* (10.1), *Lipid oxidation* (10.0) and *Response to endoplasmic reticulum stress* (10.0).

In the Cellular Component category, 15 GO terms were significantly enriched among the differential proteins (S5 Table), which were grouped into 4 groups of functional related GO terms (summarized in Fig 3B); the groups with the highest enrichment scores belonged to *Cytosolic part* (82.1), *Stromule* (22.8), *Nucleosome* (14.8) and *Photosystem* (14.1).

Furthermore, the enrichment analysis uncovered the most representative KEGG or REACTOME biological pathways in this dataset of the 80 differential proteins, which are listed in Table 3. Comparing the results from KEGG and REACTOME pathways, the protein subset is enriched in *Ribosome* and *SRP-dependent cotranslational protein targeting to membrane*, with the highest enrichment (lowest FDR) scores, respectively, and *Glycolysis/Gluconeogenesis* and *Glucose metabolism* with the lowest enrichment scores, respectively.

The significantly enriched pathway with Reactome code R-ATH: 3371497- *HSP90 chaperone cycle for steroid hormone receptors (SHR)* is very relevant in the context of this investigation and of the available bibliography. The innate immunity and plant defence in *Arabidopsis* are biological events that are associated with the biological function of heat shock protein 90–2 as a molecular chaperone, involved in RPM1-mediated resistance and component of the RPM1/RAR1/SGT1 complex [51]. To circumvent the autoimmunity associated with high levels of immunity receptors, HSP90 proteins may assist in the formation of protein complexes that target the immune receptors SNC1, RPS2, and RPS4 for degradation [52].

## Protein abundance patterns associated with GO functional categories

**GO biological process category.** In the enrichment analysis, 18 proteins contributed significantly to certain biological processes (GO_BP groups of enriched GO_BP terms), which are detailed in Table 4. Of the six highlighted GO-BP groups, three stand out based on the constant patterns of variation in the abundance of the associated proteins. These are: *Cellular metabolic compound salvage* (GO group 3), with four down-accumulated proteins in the inoculated plants; *Protein-DNA complex assembly* (GO group 5), with four up-accumulated proteins in the inoculated plants and *Response to endoplasmic reticulum stress* (GO group 2), with two up-accumulated proteins in the inoculated plants. In the GO_BP groups *Pyridine-containing compound metabolic process*, *Monosaccharide catabolic process* and *Lipid oxidation*, proteins with different variation forms were housed in the same group.

Within the proteins mapping to *Cellular metabolic compound salvage* (GO group 3), the protein Ribulose 1,5-bisphosphate carboxylase/oxygenase (RubisCO) small subunit 1A (P10795; RBCS1A) was the one showing the most expressive negative variation between inoculated and control samples (Log2FC = -3.1). Further, this protein showed a very high coefficient of variation (% CV), both in control (151%) and inoculated samples (152%). CV values may

**Table 3. Biological pathways uncovered for the selected dataset.**

| ID | Term | Source | Term FDR | Group FDR | Enrichment score | Groups | % Associated proteins | Number of proteins |
|---|---|---|---|---|---|---|---|---|
| KEGG:03010 | Ribosome | KEGG | 6.46E-08 | 3.69E-08 | 24.7 | 4 | 5.49 | 20.0 |
| KEGG:00630 | Glyoxylate and dicarboxylate metabolism | KEGG | 1.20E-03 | 6.86E-04 | 10.5 | 2 | 8.00 | 6.0 |
| KEGG:00710 | Carbon fixation in photosynthetic organisms | KEGG | 4.03E-03 | 2.30E-03 | 8.8 | 3 | 7.25 | 5.0 |
| KEGG:00010 | Glycolysis / Gluconeogenesis | KEGG | 2.64E-02 | 1.51E-02 | 6.1 | 1 | 4.35 | 5.0 |
| R-ATH:1799339 | SRP-dependent cotranslational protein targeting to membrane | REACTOME | 4.98E-10 | 3.98E-08 | 24.6 | 3 | 7.39 | 17.0 |
| R-ATH:3371497 | HSP90 chaperone cycle for steroid hormone receptors (SHR) | REACTOME | 5.16E-05 | 1.06E-02 | 6.6 | 2 | 22.22 | 4.0 |
| R-ATH:70326 | Glucose metabolism | REACTOME | 2.99E-02 | 2.99E-02 | 5.1 | 1 | 4.76 | 3.0 |

**Table 4. Proteins from the selected dataset assigned to Biological process GO terms with the highest enrichment scores.**

| Enrichment analysis | | SWATH analysis | | Data analysis | | |
|---|---|---|---|---|---|---|
| GO Group title and number | GO group[a] | *Arabidopsis* Uniprot Accession[b] | Protein name (initials)[b] | Potential subcellular location[c] | Potential pathway or biological processes[d] | LOG2FC[e] |
| Pyridine-containing compound metabolic process (4) | 4.6 | O04499 | 2,3-bisphosphoglycerate-independent phosphoglycerate mutase 1 (PMG1/iPGAM) | Cytoplasm | Glycolysis | 0.9 |
| | 4 | Q9SRZ6 | Isocitrate dehydrogenase [NADP] (ICDHC/cICDH) | Cytoplasm | Plant defense; Oxidative stress | 0.4 |
| | 4 | Q9LF98 | Fructose-bisphosphate aldolase 8 (ALFC8/FBA8) | Cytoplasm | Glycolysis; Stress signalling | -0.3 |
| Monosaccharide catabolic process (6) | 6 | O49299 | Probable phosphoglucomutase, cytoplasmic 1 (PGMC1/PGM1) | Cytoplasm | Carbohydrate metabolism | 0.2 |
| | 4.6 | Q42525 | Hexokinase-1 (HXK1) | Cytoplasm Nucleous | Glycolysis; Stress signalling | -0.3 |
| | 4.6 | Q9SAU2 | D-ribulose-5-phosphate-3-epimerase (Q9SAU2/RPE) | Chloroplast | Photosynthesis | -0.2 |
| Cellular metabolic compound salvage (3) | 3 | P10795 | Ribulose bisphosphate carboxylase small chain 1A (RBS1A/RBCS1A) | Chloroplast | Photorespiration; Photosynthesis | -3.1 |
| | 3 | P25697 | Phosphoribulokinase (KPPR/PRK) | Chloroplast | Photosynthesis; Plant defense | -0.6 |
| | 3 | Q9LRR9 | (S)-2-hydroxy-acid oxidase GLO1 (GLO1/GOX1) | Peroxisome | Plant defense; Photorespiration | -0.6 |
| | 3.1 | F4KDZ4 | Malate dehydrogenase (F4KDZ4/PMDH2) | Peroxisome | Fatty acid ß-oxidation | -0.3 |
| Protein-DNA complex assembly (5) | 5 | A8MRL0 | Histone superfamily protein H3.3 (A8MRL0/AT4G40030) | Nucleus | DNA-binding; Protein heterodimerization | 1.1 |
| | 5.2 | Q9SEI3 | 26S proteasome regulatory subunit 10B homolog A (PS10A/RTP4A) | Nucleus | Effector Triggered Imunity; ATPase activity | 0.6 |
| | 5 | P59259 | Histone H4 (H4/HIS4) | Nucleus | Nucleosome assembly; Protein heterodimerization | 0.5 |
| | 5 | Q8LB10 | ATP-dependent Clp protease proteolytic subunit-related protein 4 (CLPR4) | Chloroplast | Plastid protein homeostasis; Protein degradation | 0.3 |
| Lipid oxidation (1) | 1 | P38418 | Lipoxygenase 2 (LOX2) | Chloroplast | Lipid metabolism; Biotic stress | 1.2 |
| | 1 | F4JYM8 | Thiolase family protein (F4JYM8/AACT1) | Peroxisome | Transferase activity | -0.6 |
| Response to endoplasmic reticulum stress (2) | 2 | P27323 | Heat shock protein 90–1 (HSP901) | Cytoplasm | Chaperone; Plant defense | 1.7 |
| | 2 | Q9S9N1 | Heat shock 70–5 (HSP7E/BiP1) | Endoplasmic Reticulun | Chaperone; Plant defense | 0.7 |

[a]The proteins included in more than one GO group were referenced only once in Table 4 with an indication of the numbers of the groups with which they were associated.

[b]The table includes information about the protein names linked to *Arabidopis* Uniprot Accessions, used as a reference for the cork oak leaf proteomic profiles.

[c]Suggestions of the potential subcellular locations, based on the annotation in protein databases and available bibliography.

[d]Inferences about the possible biological processes associated with proteins in the context of this study.

[e]Fold change ratio logarithm of protein abundance in inoculated samples over control greater than 1 or less than -1 (Log2FC).

reflect the natural biological variability observed for RubisCO in *Q. suber* and the corresponding RubisCO patchiness in the host's response to the oomycete. In *Quercus ilex* inoculated with *P. cinnamomi* a decrease in the abundance of RubisCO proteins was also found, which was correlated with the tolerance/susceptibility of the provenances, being more accentuated in susceptible provenances [53].

RubisCO is very abundant in plants and the amount of this protein in the leaves is considered an indicator of the photosynthetic vigour and nitrogen availability. RBCS1A is a member of the multigene family RBCS from *Arabidopsis* and Isumy and colleagues [54] reported the additive effect of the expression of RBCS1A and RBCS3C genes on RubisCO accumulation in *Arabidopsis* leaves. Therefore, low levels of RBCS1A in the *P. cinnamomi* inoculated cork oak plants may indicate decreased levels of total RBCS mRNA and a smaller content of RubisCO accumulated in the leaves. It will be interesting to evaluate if the leaves of cork oak plants inoculated with *P. cinnamomi* have a reduced photosynthetic activity but, if this was a decrease, it appears not to have significantly affected plant growth that was similar between inoculated and control plants.

Like RubisCO, the chloroplastic phosphoribulokinase (P25697/PRK) is specifically associated with the Calvin-Benson cycle and catalyses D-ribulose 1,5-bisphosphate formation, used by RubisCO with $CO_2$ or $O_2$ to form 3-phosphoglycerate (3-PGA) and 2-phosphoglycolate (2-PG) [55,56]. Similarly, this protein also showed lower levels of accumulation in the *P. cinnamomi* inoculated plants in this study. The protein D-ribulose-5-phosphate-3-epimerase protein (Q9SAU2/RPE), mapping to GO groups 4 and 6 and participating in the carbon photoassimilation cycle, also showed lower levels of accumulation in the inoculated plants like RubisCO and PRK. It is likely that the decrease in the accumulation of these proteins can compromise the levels of carbon assimilated by the plant and the sequent synthesis of sugars, proteins, lipids and nucleic acids. A similar effect was observed in *Q. ilex* inoculated with *P. cinnamomi* where many proteins involved in the Calvin-Benson cycle, such as RubisCO large and small subunits, phosphoglycerate kinase, glyceraldehyde-3-phosphate dehydrogenase B and transketolase 1 were also decreased [53]. The complementarity of these data makes sense if we think about the existence of macromolecular complexes formed by phosphoribulokinase and glyceraldehyde-3-phosphate dehydrogenase interacting with the small peptide CP12, with relevance to the regulation of photosynthesis in the chloroplasts [57,58].

The two other proteins mapping to this GO group, (S)-2-hydroxy-acid oxidase, with the alternative name of glycolate oxidase 1 (Q9LRR9; GOX1/GLO1) and malate dehydrogenase (F4KDZ4; PMDH2), are known to be located in the peroxisomes, and both showed decreased levels in the inoculated plants. GOX1 catalyses the conversion of glycolate into glyoxylate with the production of $H_2O_2$ in the photorespiration pathway (EC 1.1.3.15). Modulation of hydrogen peroxide accumulation has been suggested as the mechanism adopted by the GOX protein family in *Arabidopsis* and *Nicotiana benthamiana*, associated with PAMP-triggered immunity (PTI), host and nonhost defence responses [59–61]. Furthermore, the defence pathways activated by different GOX genes vary between plant species and depend on the type of interaction that occurs between plants and pathogens or elicitors and GOX1-dependent defence responses may involve salicylic acid (SA) and *WRKY62*-mediated pathways [59–61].

Concerning malate dehydrogenase (F4KDZ4; PMDH2), in 2007, Pracharoenwattana and colleagues proposed a model in which the action of this enzyme is the production of malate from oxaloacetate with NADH oxidation, recruited for fatty acid ß-oxidation.

In summary, the decrease in the accumulation of these four proteins rebound on photosynthesis and concomitant photorespiration, and may affect sugar metabolism. Triacylglycerides oxidation may be used as an alternative source of energy and supply of gluconeogenic intermediates. Changes in the redox state of the cells resulting from the production of reactive oxygen species (ROS) are perceived and result in the activation of the defence system. Compromising the photosynthetic efficiency in source tissues may result in a reduced supply of sugars to sink tissues and less accumulation of soluble sugars.

Focusing on the *Pyridine-containing compound metabolic process* (GO group 4) and *Monosaccharide catabolic process* (GO group 6), hexose sugars like glucose are central molecules in

plant metabolism and in sugar signaling. Cytosolic resources of phosphate hexoses originating from starch mobilization and sucrose hydrolysis are channeled for energy-producing and synthesis of biomolecule precursors [62]. Proteins PGMC1/PGM1 (Probable phosphoglucomutase), PMG1/iPGAM1 (2,3-bisphosphoglycerate-independent phosphoglycerate mutase 1), ALFC8/FBA8 (fructose-bisphosphate aldolase 8) and HXK1 (Hexokinase 1) from GO groups 4 and 6 can be grouped into two pairs according to their quantification pattern but also to the role they play in sugar metabolism and as sugar sensors. PGMC1/PGM1 is an enzyme that participates in both the breakdown and synthesis of glucose (EC: 5.4.2.2.) and PMG1/iPGAM1 is involved in the synthesis of pyruvate in glycolysis (EC: 5.4.2.12). These proteins were more abundant in the inoculated samples of this study, revealing a metabolic tendency *in favour* of energy production and reducing power as opposed to the accumulation of sucrose and carbohydrates as reserve substances. The availability of energy resulting from the functioning of enzymes may hamper energy-depending cell actions, such as the movement of stomata, and this requirement was studied in *Arabidopsis* through silencing the expression of glycolytic proteins. Silencing *iPGAM* activity in *Arabidopsis* is associated with reduced stomatal function and plant phenotypes with delayed development. This, probably results from the decrease in ATP production by the glycolytic pathway and also by tricarboxylic acid (TCA) cycle and oxidative phosphorylation in consequence of the concomitant reduction in the levels of pyruvate provided [63]. HXK1 and FBA8 were less abundant in the inoculated samples and both enzymes are involved in glycolysis (EC: 2.7.1.1; EC: 4.1.2.13) and sucrose metabolism, also being referred to as proteins involved in sugar and stress signaling [64,65]. In *Arabidopsis*, transcripts levels of *AtFBA8* showed increased expression after 24h of glucose, fructose and sucrose treatment and these were responsive to ABA, SA, NaCl and drought stresses [65]. A reduction in the production of these enzymes in the inoculated cork oak plants may be a consequence of the imbalance of the metabolism towards the production of energy associated with the immune response. Knowing the subcellular location of HXK1 is essential to understand the role it plays in response to biotic stresses, because HXK1 located in the nucleus may interact with other proteins regulating the transcription of genes by binding directly to the chromatin and mitochondrial hexokinases can modulate programmed cell death (PCD) [66].

Cytosolic ICDHC/cICDH (Isocitrate dehydrogenase [NADP]) protein, mapping to GO group 4 and increasing its levels in inoculated plants, is potentially responsible for 2-oxoglutarate production for amino acid biosynthesis; however, in *Arabidopsis* cICDH is not required for plant development and primary metabolism in optimal growth conditions, instead, cICDH contributes to thiol–disulphide homeostasis during oxidative stress [67]. The NADPH produced by cICDH may contribute to activate defence responses to pathogen infection that are triggered by changes in cellular redox state [67].

Reprogramming gene expression in situations of biotic stress requires modulation of transcriptional activity in the nucleus. Focusing now on *Protein-DNA complex assembly* (GO group 5) and *Response to endoplasmic reticulum stress* (GO group 2), several proteins were found with increased levels, including two histones. It was suggested that appropriate levels of H3.3 are required to avoid H1 deposition over gene bodies preserving an adequate density of nucleosomes ideal for chromatin unfolding and access to DNA methyltransferases that methylate gene bodies [68]. Besides, the local enrichment of the nuclear histone H3.3 variant was positively correlated with transcription of responsive genes [69,70] and with gene body methylation [68]. Thus, the higher levels of H3.3 protein in the inoculated plants in this study suggest an increased access to DNA, allowing for modulation of transcription of biotic responsive-genes through gene body methylation. Still related to the formation of histone-DNA tetrasome is the *Arabidopis* chaperone NASP, described to bind to H3-H4 dimers and to stimulate the conversion of dimers to tetramers, *in vitro* [71]. Furthermore, in tobacco and

*Arabidopsis* cell lines the modifications observed in histone H3 in response to abiotic stresses, with up-regulation of marker genes, happens together with histone H4 acetylation, revealing the parallel intervention of these histones [72]. Also, in the present study, a significant up-accumulation of both H3.3 and H4 proteins were found in the leaves of cork oak plants inoculated with *P. cinnamomi*.

To reach cellular homeostasis, protein degradation by proteolytic enzymes is a regulated procedure used to adjust protein abundance and efficiency. This activity in the chloroplasts requires Clp protease complexes composed by several protein catalytic (ClpP3 to ClpP6) and non-catalytic (ClpR1 to ClpR4) subunits arranged in ring-like structures (P-ring and R-ring) in *Arabidopsis* surrounding the proteolytic chamber whose activity is assisted by several chaperone members [for review see 73]. The assembly of the rings of the Clp core complex is compromised if there is an uneven number of subunits available for its formation. It was observed that reducing the abundance of the subunit ClpP6 by 50% caused a reduction in the protein abundance of other P- and R-ring components, interfering with the complex assembly and functionality [74]. Therefore, it is reasonable to expect that the increase in accumulation of the ClpR4 subunit in the inoculated cork oak samples, may point to the importance of protein degradation in the regulation of the photosynthetic process mediated by the Clp complex. Additionally, knockdown of protease subunits in tobacco allowed the identification of putative protease substrates, including proteins involved in photosynthesis like PRK and RPE, which were found to be down-accumulated in the cork oak samples inoculated with *P. cinnamomi* (Table 4).

Beyond the Clp complex, there are other biological mechanisms that predict protein degradation, through proteasome complexes, to modulate the activity of disease resistance proteins (R) in plant-pathogen interactions and also other processes such as the oxidative burst, hormone signaling, gene induction, and programmed cell death [75]. In tobacco cells challenged by the elicitin cryptogein, the accumulation of 20S proteasome subunits was observed simultaneously with the development of systemic acquired resistance [76]. By analogy, the RPT4A protein highlighted in the leaf proteome of the inoculated samples in the present study can be regarded as a defence-induced subunit of 26S proteasome, with a possible role in plant defence reactions eventually triggered by elicitins produced by *P. cinnamomi*. It is also possible to assume ATPase activity for *Q. suber* RTP4A subunit based on the functional characterization of the 26S proteasomal subunit RPT4a from *Solanum lycopersicum* that has an active ATPase site and can modulate the resistance to the ToLCNDV virus by physically interacting with viral DNA molecules [77].

Modifying the programming of host nuclear gene transcription in response to biotic stress is one of the mechanisms adopted by oomycetes and promoted through effector molecules. The target genes may be those that code for HSP (Heat Shock Proteins) proteins, modulating the role played by these molecular chaperones; these are active partners of numerous enzyme complexes and are responsible for the folding and unfolding of proteins included in protein degradation/renaturation and movement of signaling proteins and transcription factors into cell organelles [78,79]. In *Q. suber* leaves inoculated with *P. cinnamomi*, the accumulation of HSP70-5 (Q9S9N1) and HSP90-1 (P27323) proteins was higher than in control samples. Knowing the activities of these proteins in other plant species it can be inferred the possible role they may play in the interaction between cork oak and *P. cinnamomi*. Song et al (2015) reported the identification of a *P. sojae* intracellular CRN (Crinkler or crinkling- and necrosis-inducing protein) effector which directly interacts with promoters of the genes encoding HSP proteins, preventing the binding of specific transcription factors [80]. The expression of the defence-related genes in *Arabidopsis*, *N. benthamiana* and soybean is then changed, unbalancing the host's resistance level to *Phytophthora* species [80]. More recently, HSP70s have been

noted as proteins that interact with RXLR effectors produced by *P. infestans* and that get involved in *N. benthamiana* defence response by stimulating programmed cell death mediated by MAPK signaling and suppressing the growth of the pathogen [81]. It was also reported the importance of protein complexes formed between HSP90s and co-chaperones in the activation of defence mechanisms mediated by resistance (R) proteins after the detection of pathogen effector molecules [82]. Preventing the formation of these protein complexes by inhibiting the binding of HSP90 has implications for the accumulation of R proteins and resistance mediated by these proteins [86]. In *Arabidopsis*, the cytosolic AtHSP90.1 was the only HSP90 isoform significantly induced after inoculation with *Pseudomonas* (*Pst*) strains containing avirulence genes (*avrRpm1* and *avrRpt2*) and was required for the full resistance mediated by one of the corresponding R proteins [83]. It is then expected that the host's resistance proteins will recognize the effector molecules secreted by *P. cinnamomi* and activate the defence responses with the collaboration of chaperones and co-chaperones.

The disclosed *Q. suber* HSP70-5 is recognized as a homolog of the *Arabidopsis* BIP1 for Binding Immunoglobulin Protein or Binding Protein (BiP), a chaperone set in the endoplasmic reticulum (ER) lumen, known to bind a membrane-associated transcription factor (TF) under non-stressed conditions [84]. The interaction between BiP1 and the TF is the requirement to turn on or off a protein secretory signaling pathway via ER-Golgi-Nucleus, ending with the transcription of stress response genes [84]. In soybean, the BiP protein was described as a negative regulator of a stress-induced cell death response and, in *Arabidopsis*, Wang et al. (2005) reported the implications of *BiP 2* silencing on the secretion of pathogenesis related-proteins, compromising the systemic acquired resistance against bacterial pathogens [85].

When looking at the enrichment of *Lipid oxidation* (GO group 1), two differential proteins were assigned to it, lipoxygenase 2 (LOX2; P38418) and thiolase family protein (AACT1; F4JYM8), although with different patterns of variation: LOX2 was more abundant in the inoculated plants and AACT1 was less abundant. By homology to the *Arabidopsis* ACCT1 isoform, it is expected for *Q. suber* ACT1 to be located in the peroxisome, based on the presence of two alternative targeting sequences PTS1 and PTS2 motives found in AtACCT1, excluding a metabolic function related to isoprenoid biosynthesis [86,87]. Jin et al (2012) [88] found a strong expression of *AtACCT1* in the vascular system of the aerial organs and roots of *Arabidopsis*, verifying that gene silencing or induction of abiotic stresses did not result in an evident phenotypic response [88]. In *Q. suber* inoculated by *P. cinnamomi*, the leaf proteome reveals a reduction in the production of a thiolase AACT1 protein, apparently included in its defence strategy.

The involvement of LOX2 in the cork oak defence response may be associated with the production of jasmonic acid (JA) via the Vick and Zimmerman pathway [89]. Recently, it was confirmed that LOX2 forms a protein complex with AOS (allene oxide synthase) and AOC2 (allene oxide cyclase), two proteins that also participate in JA precursor biosynthesis, located in the inner envelope of the *Arabidopsis* chloroplasts [90]. The formation of this molecular complex is evident in the effectiveness of JA production to the disadvantage of other products resulting from parallel reactions during oxylipin biosynthesis, guiding the defence response to the activation of genes responsive to JA [90]. Sometimes, depending on the needs of the pathogen, the signaling reactions are a balance between the activation of the salicylic acid (SA) pathway with suppression of the jasmonic acid (JA) pathway or vice versa [91]. Nevertheless, Starý et al. (2019) conclude that the level of resistance induced in different tomato genotypes after β-cryptogein treatment correlated with the upregulation of defence genes and activated ethylene and JA signaling but not SA signaling [92]. Other authors refer to a biphasic defence response in avocado against the hemibiotroph *P. cinnamomi*, which initially involves SA-mediated gene expression followed by the enrichment of JA-mediated defence from 18 to 24 hours post-inoculation [93].

Finally, when analysing the GO Cellular Component category, the highest enrichment scores were obtained for *Cytosolic part* (82.1), *Stromule* (22.8), *Nucleosome* (14.8) and *Photosystem* (14.1). In a simplified view, it is recognized that achieving new cell balances during interaction with pathogenic organisms requires the remodeling of physiological processes by the action of cytoplasmic or organelle-associated enzymes and the modulation of transcription factors for nuclear gene expression, which are energy-dependent processes. Three of the obtained categories fit this profile with the exception of *Stromule*, which is a novelty for the host-*P. cinnamomi* interactions. Nevertheless, there were previous descriptions for the importance of the communication between cellular organelles during immune responses carried out through stroma-filled tubular structures (stromules) of the chloroplasts-to-nucleus, which use them as a support for the exchange of molecules integrated into the defence response [94,95]. Caplan et al (2015) observed the induction of stromules in response to viral and bacterial effectors after recognition by host receptors (ETI; effector-triggered immunity) [94]. In plants infected with tobacco mosaic virus (TMV) a hypersensitive response is observed at the infection site and in the border regions with increased production of stromules in both areas, probably stimulated by the production of pro-defence signaling molecules like $H_2O_2$, $O_2^-$ and SA [95]. During innate immunity, the cellular relocation of chloroplasts in the nucleus surroundings is dependent on the organization of microtubules in connection with the anchoring points provided by actin filaments to enhance the effectiveness of the communication between these organelles [96]. In 2013, Sghaier-Hammami et al. reported the up-accumulation of actin in holm oak plants inoculated with *P. cinnamomi* [53]. The published information reinforces the importance of the *Stromules* GO category highlighted in the present study for the cork oak-*P. cinnamomi* interaction. Moreover, the cork oak leaf proteome data suggests a possible function for stromules in long-term defence responses, far from the inoculation point, in close connection with the production and transport of signaling molecules.

## Conclusions

In this work, the proteomes of cork oaks plants submitted to biotic stress-induced by *P. cinnamomi* inoculation are revealed for the first time. Among the 424 proteins confidently quantified in the inoculated and non-inoculated plants, a dataset of 80 proteins was selected based on the abundance variability observed between the experimental conditions. The immune response of the plants was analysed eight months after the inoculation event, and, at that moment, there were no evident phenotypic differences between inoculated and non-inoculated plants. Nevertheless, the hierarchical clustering of differentially produced cork oak proteins shows two different groups of plants, matching to the experimental conditions. By comparing protein profiles, it was observed that the number of proteins in which the abundance increased in the inoculated plants is 3 times greater than the number of proteins in which there was a decrease in abundance. Therefore, the defence responses induced in the host by *P. cinnamomi* in the long term and distant from the inoculation site are inscribed in the proteome of the leaves, reproducing the in progress homeostatic state of the plants. The results obtained in this study increase the possibilities of screening trees infected with *P. cinnamomi* using protein markers identified in the leaves without the need to isolate the oomycete from the roots of the host or surrounding soil.

The homeostatic state of the inoculated cork oak plants was characterized by protein patterns associated with differential biological processes occurring potentially in different subcellular organelles. When performing the enrichment analysis, eighteen proteins were highlighted, and their possible functions in an immune response context were discussed. In short, the decrease in the accumulation of photosynthesis enzymes and concomitant

photorespiration may compromise the levels of carbon assimilated by the plant and its development, although throughout the experiment, no differences in growth were observed. The dynamics of proteins associated with sugar metabolism and sugar signaling reveals a metabolic tendency *in favour* of energy production and reducing power as opposed to the accumulation of sucrose and carbohydrates as reserve substances.

The reprogramming of gene expression, eventually in response to the action of effector molecules produced by *P. cinnamomi* is a major function associated with proteins that are in greater abundance in the inoculated plants. It is also clear the participation of proteolytic complexes and chaperones in the cork oak immune response and of proteins sensitive to changes in the redox state of the cell promoted by ROS species.

In addition, the cork oak leaf proteome data suggests the importance of the communication between cellular organelles mediated by stromules in the long-term defence responses. Immune response amplification and effectiveness may be dependent on the repositioning of the chloroplasts close to the nucleus and the transfer of pro-defence molecules such as SA, JA and $H_2O_2$.

## Supporting information

**S1 Fig. Biological material.** Information on biological material and procedures performed in the experimental assay.
(TIF)

**S2 Fig. Visual observation of the plants over the duration of the experiment.**
(TIF)

**S1 Table. Cork oak references.** Plant labels and GPS references for location of cork oak parental trees.
(PDF)

**S2 Table. Proteins identified and quantified in cork oak leaves compared to the *Arabidopsis thaliana* reference proteome.** List of the 608 proteins identified by IDA analysis (Table S2.1.) and list of the 424 proteins quantified by SWATH-MS in control (C) or in inoculated (I) cork oak leaves (Table S2.2.).
(XLSX)

**S3 Table. Proteins identified and quantified in cork oak leaves compared to the proteins deduced from the draft genome of cork oak.** List of the 1388 proteins identified by IDA analysis (Table S3.1.) and list of the 841 proteins quantified by SWATH-MS in control (C) or in inoculated (I) cork oak leaves (Table S3.2.).
(XLSX)

**S4 Table. Significantly enriched GO biological process terms and groups in the list of 80 differential proteins.**
(PDF)

**S5 Table. Significantly enriched GO cellular component terms and groups in the list of 80 differential proteins.**
(PDF)

**S6 Table. Significantly enriched GO biological process groups in the list of 80 differential proteins.**
(PDF)

**S7 Table. Significantly enriched GO cellular component groups in the list of 80 differential proteins.**
(PDF)

**S1 File. Description of the SWATH-MS.** Principles and detailed materials and methods.
(PDF)

## Author Contributions

**Conceptualization:** Ana Cristina Coelho.

**Data curation:** Ana Cristina Coelho, Gabriela Schütz, Cátia Santa, Bruno Manadas, Patrícia Pinto.

**Formal analysis:** Gabriela Schütz, Cátia Santa, Bruno Manadas, Patrícia Pinto.

**Funding acquisition:** Ana Cristina Coelho, Cátia Santa, Bruno Manadas.

**Investigation:** Ana Cristina Coelho, Rosa Pires.

**Methodology:** Ana Cristina Coelho.

**Project administration:** Ana Cristina Coelho.

**Resources:** Ana Cristina Coelho.

**Supervision:** Ana Cristina Coelho.

**Visualization:** Ana Cristina Coelho.

**Writing – original draft:** Ana Cristina Coelho.

**Writing – review & editing:** Rosa Pires, Gabriela Schütz, Cátia Santa, Bruno Manadas, Patrícia Pinto.

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
