## [Decision Letter · Decision Letter 0]

20 Oct 2020

PONE-D-20-29356

Disclosing proteins in cork oak plants associated with the immune response to Phytophthora cinnamomi inoculation in a long-term assay

PLOS ONE

Dear Dr. Coelho,

Thank you for submitting your manuscript to PLOS ONE. After careful consideration, we feel that it has merit but does not fully meet PLOS ONE’s publication criteria as it currently stands. Therefore, we invite you to submit a revised version of the manuscript that addresses the points raised during the review process.

The ms «Disclosing proteins in cork oak plants associated with the immune response to Phytophthora cinnamomi inoculation in a long-term assay» was reviewed by four specialists who classified the ms for major revision and added comments, suggestions, corrections, which can assist the authors to prepare an improved version of the ms.

Title could mention the main clue of the paper, «distal analysis»;

Introduction

I agree that the new version could be improved by removing redundant text; however the reviewers call your attention to the importance of describing SWATH-MS quantitative proteomics;

Materials and Methods

This section needs your best attention: sample size due to the absence of biological replicates;  experimental design; namely root analysis only at 24 and 48h;  confident reproducibility of results; use of cork oak genome database; better phenotyping including physiological parameters because the difference in proteomics betrays differences in physiological parameters;

Results

Please pay attention to the reviewer´s comments about quality and redundancy of figures; figures and tables in ms  main body  or as supplementary material.

Please submit your revised manuscript not later than 2020 dec 20th. If you will need more time than this to complete your revisions, please reply to this message or contact the journal office at plosone@plos.org. Please include the following items when submitting your revised manuscript:

We look forward to receiving your revised manuscript.

Kind regards,

Sara Amancio

Academic Editor

PLOS ONE

Journal Requirements:

Reviewers' comments:

Reviewer's Responses to Questions

**Comments to the Author**

1. Is the manuscript technically sound, and do the data support the conclusions?

Reviewer #1: Partly

Reviewer #2: No

Reviewer #3: No

Reviewer #4: Yes

2. Has the statistical analysis been performed appropriately and rigorously? 

Reviewer #1: I Don't Know

Reviewer #2: I Don't Know

Reviewer #3: I Don't Know

Reviewer #4: Yes

3. Have the authors made all data underlying the findings in their manuscript fully available?

Reviewer #1: Yes

Reviewer #2: No

Reviewer #3: Yes

Reviewer #4: Yes

4. Is the manuscript presented in an intelligible fashion and written in standard English?

Reviewer #1: No

Reviewer #2: Yes

Reviewer #3: No

Reviewer #4: No

5. Review Comments to the Author

Reviewer #1: The research topic is very interesting; nevertheless I think the experimental design has important flaws:

1 –The sample size is not adequate. Only 6 plants were inoculated with P. cinnamomi and 6 were not inoculated. Due to the fact that the plant material is very heterogeneous, originated from seeds, making each individual a unique genotype, and due to the high heterozigoty of the species Quercus suber it was necessary to have a bigger sample size to increase the robustness of the assay and obtain more reliable results;

2 – As the plants haven't died 7 months after inoculation, the virulence of the isolate is questioned. How this isolate was selected? Did the plants die in the previous virulence assay with PA 45 strain?

3 – Although the authors hypothesis is that after inoculation of plant roots with a pathogen, an immune response is initiated that will lead to a new homeostatic state, with protein changes that can be detectable in the long-term, distally from the infection site, I think that it was important also to compare the proteome of the roots (the site of infection) with the proteome of leaves. But this should be done with biological replicates. Biological replicates are also an important issue in these studies and the lack of biological replicates can be considered a problem with this experiment. Due to the difficulty of having biological replicates for this species, a bigger sample size would be necessary.

4 – Why the experiment lasted for 248 days?

Reviewer #2: Dear Authors,

I have the opportunity to revise your ms on Q. suber-P .cinnamomi interaction. I find the subject highly relevant and of interest for the community. In my view, the analysis of the leaf proteome evaluation of root infection and the long term analysis (> 200 days after inoculation) are particularly interesting.

The proteomic approach is well described, exceptions being the designation of the particular method used and the absence of Q. suber genome database. The meaning of SWATH-MS only appears in lines 233-234 and while authors sustain it is a novel approach (e.g. lines 164-165) its benefits and main features are not explained. Researches in the proteomic field will be able to navigate through this but others will struggle. Authors need to explain why not use the Q. suber genome database.

However, the comprehensive proteomics work is not supported by a physiological assay, including the success and extension of infection and if mortality was observed. A single photo is presented, in which the rot is mentioned but not very visible.

So, a large part of the biological conclusions, including marker proteins, are unsupported by the presented data. In my view, the publication of the authors’ findings, and their hypothesis about processes trigged by infection requires an extensive revision and the inclusion of such data.

Authors indicate the two groups of plants are not distinguished at the end of the assay. However, the data strongly support s for altered primary metabolism including carbon assimilation. This highlights the need to clearly explain which type of leaf was used for analysis.

Two acorns per tree were used with a total of six trees. Can authors provide more info about the parentals? Were seeds taken from a natural regeneration stand? How likely it is they came from the same gene pool? In addition, as the parental were showing evidence of decline other questions arise:

a) Which are the causes of the decline?

b) Was the extension of damage similar?

c) How was priming effects addressed in the study?

d) Were only two seed per parental used?

e) Given each parental source, was the kinetics of infection, the amount and severity of symptoms similar?

At this point, and after analysing Introduction, M&M and results, I recommend extensive revision. Please try to keep your sentences short.

Some detailed comments:

Line 62. Is Portugal the only county affected by cork oak decline?

Line 69 and following. In the ms, the type of infection (biotrophic, hemibiotrophic or necrotrophic) is absent as well as the typical immune plant responses it. In my view, it is important to describe the system and will help readers to better follow the discussion.

Line 86. RxLr domain of?

Lines 87-90. Reprogramming in what way? Are HR and PCD promoted upon infection? If I recall properly, higher PDC and HR are typical of plant resistance responses.

Line 91. Please explain briefly compatible and incompatible interactions. Please add more info about the known mechanisms in your system. What happens in your dataset? If info is inexistent or contradictory, please add such info.

Lines 91-100. Please make this section more clear. In its current version, PTI and ETI strategies are mixed.

Lines 133-136. Relevance?

Line 155. As authors make leaf extracts, how is this approach non-invasive?

Lines 164-165. Relevance?

Lines 173. Please described which type of leaf was used and if they were asymptomatic. Add the methodology used to quantify it or score it. It would be important to have leaf characterization as leaf age, area or biomass.

Line 181. Please provide GPS coordinates.

Line 183. Distinct developments stages. Such as?

Line 185. Please provide strain characteristics.

Lines 197-199. How was the success of infection established? Please provide details on the soil characteristics and soil nutrition as mineral availability has an impact on the oomycete.

Line 237, line 257 & S1 file. Authors should run the data on a more recent database. The dbases were used more than two years before submission.

File S1. Please explain why to exclude peptides with biological modifications. What is a biological modification, a PTM?

Line 241. Per plant, eight replicate MS-runs, i.e. 2 groups x 6 plants x 8 replicate runs?

Lines 305-307. Please add the quantitative data or scoring matrix.

Line 307-309. How authors deal with this issue and how it impacts the findings of this ms?

Line 326-327. How does this variability relate to the ones found in other studies? In my view, the sentence is too simplistic and needs revision. Several factors can support such differences.

Line 280-1. Was infection at this point confirmed? Biomass and growth parameters are highly relevant for all the remaining discussion.

Lines 293-297. The info needs to be presented much earlier in the ms.

Line 328-329. Something seems to be missing in the sentence. Please confirm.

Lines 391-395. This part is not clear to me (please consider previous comments). Does homeostatic state mean there is an on-going infection?

Lines 396-398. This part is not clear to me.

Lines 398-400. Idem.

Lines 441. Highest and lowest scores mean?

Lines 464-465. The sentence is not clear to me.

Lines 493-494. Your observation made me wonder about the values of the protein for the same parental origin. Could parental origin be a factor for the difference?

Lines 503. RBCS1S and RBCS3C are both from At. Please confirm. Also, please check for consistency in RubisCO designation (line 510, 512, elsewhere?).

Lines 508-509. A possible explanation for it?

Lines 510-520. In my view, this paragraph is not in line with the previous one.

Lines 523-527. The inclusion of this sentence at this point is not clear to me.

Throughout the discussion. Data points out for lower assimilation capacity but authors also indicate plants do not differ in growth and appearance.

Line 580. Energy production needed for?

Line 734-735. The sentence is not clear to me.

Line 736. Does long term defence imply that plants cope successfully with the infection or not?

Lines 741-742. In my view, authors need to better support the assay with physiological observations as well as cytological (see initial comments).

Line 745. I do not agree that the immune response was observed. The authors analyse the proteome.

Lines 746-747. See previous comments.

Lines 754-777. In my view, the statements are not adequate in face of the present data.

Figure 1. Is more suitable as graphical Abstract. Type of leaf should be added as well as the main results. As figure is not very informative.

Figure 2. Symptoms not visible. The scale should be added.

Figure 3. As figure is not very informative. Add it to a graphical Abstract?

Reviewer #3: I recommend and insist on using the Cork oak database instead of Arabidopsis data base to identify proteins

Besides, we have some comments concerning the

Introduction: you have written a very long introduction mentioning a several results of others researchers, it seem a part of a review, please reduce it and mention only the related information with your paper and adding a section describing the SWATH-MS quantitative proteomics used, advantage and relative works

Materials and methods: we don´t understand the importance of the mentioned information about the seeds and table 1, please rewrite this section clearly and provide table 1 as supplementary materials

Results and discussion

In the first part of this section, you are describing the effect of inoculation after 24h and 48 h, then after 7 months. You have demonstrating the roots on the two first point of time (fig. 1 a, b) however, you didn’t do with 7months. Please rewrite this part because it is very confused, and it is difficult to understand the meaning of the sentences.

Otherwise, it I cannot proceed with the rest of paper until we receive the new list of identified protein using the Q. suber data base and of course the related information.

Figures, a very poor quality of figures are provided and lack of information like the figure 3

what is the importance of figure 1? eliminate it

Reviewer #4: I really enjoyed reading this paper, and it definitely provides new insights into cork oak-P. cinnamomi interactions with respect to plant defence. The paper also has huge relevance to other P. cinnamomi- host systems and consequently will be of real interest to many Phytophthora researchers interested in a detail understanding of host defence systems. The work also illustrates the potential of looking for the presence of oomycete pathogens in the distal plants of the plant with out the need for excavating root and disturbing the hosts when trying to ascertain if symptoms are associated with Phytophthora.

I have made a number of suggestions and comments throughout the text in the text or as comments, which should be considered before final submission (see attachment - I converted to Word for ease, so some minor reformatting will be required, apologies. Some more work is required in the methods so that the work can be easily repeated by others if they so want to.

Hopefully, these suggestions and comments help?

6. PLOS authors have the option to publish the peer review history of their article (what does this mean?). If published, this will include your full peer review and any attached files.

Reviewer #1: No

Reviewer #2: No

Reviewer #3: No

Reviewer #4: No

---

## [Author Response · Author response to Decision Letter 0]

17 Dec 2020

Cover letter of Response

 Dear Editor,

 We are pleased to submit the revised version of our manuscript (MS), which we have modified taking into account all the reviewer’s and editorial comments. We thank the reviewers and editor for their constructive and helpful comments, which we feel have improved the quality and clarity of our MS. 

We provide in this letter a summary of the main changes performed in the manuscript, in line with the main questions pointed by the editor, which is followed with the specific responses to the points raised by each of the reviewers.

The manuscript is now entitled Disclosing proteins in the leaves of cork oak plants associated with the immune response to Phytophthora cinnamomi inoculation in the roots: a long-term proteomics approach, to integrate the distal analysis information as suggested by reviewer 4.

In order to better explain the principles of the novel SWATH-MS quantitative proteomics approach, its advantages and previous applications in plants, we have added a paragraph summarizing them at the end of the introduction, a new section of Introduction in the reformulated S1 File (now designated “Description of the SWATH-MS, principles and detailed materials and methods”) and six new references. In addition, the introduction has been shortened by removing unnecessary contents in order to become more focused. 

Besides the analyses of the proteomics data using the three reference proteomes extracted from the curated Uniprot database (Plant, Arabidopsis and Populus), we have also performed both IDA identification and SWATH quantification using as reference the predicted proteins deduced from the draft genome of Quercus suber, downloaded from the CorkOakDB. The results from this analysis are now included in Table 1 and we supply both identification and quantification data in the new Supplementary S3 table. However, the results of identification and quantification using the cork oak predicted proteins were of low confidence and indicated a high redundancy of the database, as illustrated in the example given below in the detailed response to reviewers 2 and 3. These analyses made clear that only the use of a curated reference proteome like that of Arabidopsis provided enough quality in terms of confident proteomics results and could allow to proceed for functional enrichment analyses and discussion of the biological relevance of differential proteins.

 We would like to emphasize that the sample size required for a proteomics screening to identify candidate biomarkers is different from that required for a widespread application and its validation. The sample size of the present study (six biological replicates, which is above or within the range of replicates usually analysed by SWATH-MS, as shown by the example references provided in the detailed responses) already reflects some of the biodiversity that characterizes the species, and these results could eventually be extended to hundreds of trees after follow-up studies of validation of the most promising target proteins or biological processes identified by proteomics screening.

 Performing additional physiological analyses was not within the scope of the objectives foreseen for this work and there is no guarantee that molecular evidences have a corresponding detectable physiological change, because under less favourable conditions the plant can use alternative biochemical mechanisms to guarantee its physiological development. 

 In the revised version of the manuscript, the materials and methods were rewritten in detail in line with the editors and reviewers’ comments and the quality of the figures was improved. We hope that the revised MS is now considered suitable for publication in PLOS ONE.

Kind regards

Ana Cristina Coelho

Response to reviewers

Reviewer#1

The research topic is very interesting; nevertheless I think the experimental design has important flaws:

1 –The sample size is not adequate. Only 6 plants were inoculated with P. cinnamomi and 6 were not inoculated. Due to the fact that the plant material is very heterogeneous, originated from seeds, making each individual a unique genotype, and due to the high heterozigoty of the species Quercus suber it was necessary to have a bigger sample size to increase the robustness of the assay and obtain more reliable results;

Response: Dear reviewer, we appreciate the revision of our manuscript and we tried to answer all the questions, in order to improve our manuscript.

We agree that more biological samples would allow the application of statistical methods for testing hypotheses, deriving estimates and predictions, analyzing correlations, factors, etc. using additional statistical methods, which were not within the scope of this work objectives. The sample size of this study don't clash to those observed for similar studies of proteomics and aimed to make a first screening for proteins with differential levels that will require posterior validation using more samples in follow-up, focused studies for specific target proteins. The global strategy was to deal with natural molecular diversity, even with a relatively small sample size, enhancing the expression of its representativeness in the protein profiles.

Same examples of studies in which the number of biological replicates were within the same range using SWATH-MS, an expensive and labour-demanding technology, are given below: 

-Study of rice germination using SWATH-MS with 3 biological replicates [1];

-Study of lead Response in Arabidopsis using SWATH-MS with 7 biological replicates [2];

-Study of nitrogen starvation in Arabidopsis using SWATH-MS with 4 biological replicates [3];

[1] Zhang, H., et al., Analysis of dynamic protein carbonylation in rice embryo during germination through AP-SWATH. Proteomics, 2016. 16(6): p. 989-1000.

[2] Zhu, F.-Y., et al., SWATH-MS Quantitative Proteomic Investigation Reveals a Role of Jasmonic Acid during Lead Response in Arabidopsis. Journal of Proteome Research, 2016. 15(10): p. 3528-3539.

[3] Zhu, F.-Y., et al., SWATH-MS quantitative proteomic investigation of nitrogen starvation in Arabidopsis reveals new aspects of plant nitrogen stress Responses. Journal of Proteomics, 2018. 187: p. 161-170.

2 – As the plants haven't died 7 months after inoculation, the virulence of the isolate is questioned. How this isolate was selected? Did the plants die in the previous virulence assay with PA45 strain?

Response: As specified in the manuscript, PA45 was isolated from the rhizosphere of cork oak trees that showed symptoms of decline in the Algarve region and its high virulence on cork oak seedlings was extensively studied [see references 4-6 below]. The histological analysis of cork oak roots colonized by PA45 revealed penetration of the epidermal and subepidermal cell layers and invasion of the cortex, hyphae growing actively within the cortical parenchyma and host cell destruction [4]. The following information now appears in the Material and Methods (1) and in the Results and Discussion (2):

(1)"PA45 was isolated from the rhizosphere of cork oak trees that showed symptoms of decline in the Algarve region and its high virulence on cork oak seedlings was extensively studied in previous studies"

(2)The virulence of the PA45 strain had been previously tested in cork oak roots, inoculated under the same conditions as in the present study for 3 days [11]. Histological studies performed on colonized root tissue demonstrated the ability of the oomycete to invade the epidermis, cortical parenchyma and vascular cylinder both inter–and intra-cellularly, and to destroy host cells [11].

Prior knowledge about the histological analysis of root colonization in an infection model similar to our study was decisive for the selection of PA45. 

With amounts of inoculum similar to those used in this assay, plant death is not observed. Plants may eventually die when infesting the soil with large amounts of inoculum and when the plants are submitted to regular flooding to favour root infection [7]. In nature, cork oak trees can show symptoms of decline for many years. The degree of canopy defoliation intensifies over time, the trees lose their vitality, cork harvest is no longer possible and trees dry up after several years of weakening. This slow decline economically devaluates the cork oak tree.

[4] Horta M, Caetano P, Medeira C, Maia I, Cravador A. Involvement of the β-cinnamomin elicitin in infection and colonisation of cork oak roots by Phytophthora cinnamomi. Eur J Plant Pathol. 2010;127(3):427-36.

[5] Horta M, Sousa N, Coelho AC, Neves D, Cravador A. In vitro and in vivo quantification of elicitin expression in Phytophthora cinnamomi. Physiol Mol Plant Pathol. 2008;73(1-3):48-57.

[6] Hardoim P, Guerra R, Rosa da Costa A, Serrano M, Sánchez M, Coelho A. Temporal metabolic profiling of the Quercus suber–Phytophthora cinnamomi system by middle‐infrared spectroscopy. For Pathol. 2016;46(2):122-33.

[7] Serrano MS, Rios P, Gonzalez M, Sanchez ME. Experimental minimum threshold for Phytophthora cinnamomi root disease expression on Quercus suber. Phytopathol Mediterr. 2015:461-4.

3 – Although the authors hypothesis is that after inoculation of plant roots with a pathogen, an immune Response is initiated that will lead to a new homeostatic state, with protein changes that can be detectable in the long-term, distally from the infection site, I think that it was important also to compare the proteome of the roots (the site of infection) with the proteome of leaves. But this should be done with biological replicates. Biological replicates are also an important issue in these studies and the lack of biological replicates can be considered a problem with this experiment. Due to the difficulty of having biological replicates for this species, a bigger sample size would be necessary.

Response: P. cinnamomi is a root rot pathogen and all studies characterizing the infection of cork oak that were previously carried out were directed at the roots and for infection times up to 72h. We highlight studies of transcriptomics [8] and metabolomics [9], as well as those related to the expression of genes involved in the cork oak defence Response to P. cinnamomi [10, 11]. We agree that it would be important to study the root proteome and comparing it with the leaf proteome for a better understanding of the molecular mechanisms of interaction between P. cinnamomi and the host. This is however beyond the scope of the current manuscript but it would be an interesting study. In addition, the approach here taken is aimed at using leaf protein markers to detect trees potentially infected with P. cinnamomic, which can be of great help in the management of cork oak forests. Cork producers face a serious economic problem with the death of the trees and ask the researchers for answers. The survey for protein markers in the leaves of adult trees in the field can be an effective and non-invasive strategy for the early diagnosis of infected trees, replacing the classical procedures for P. cinnamomi isolation from the rhizosphere of trees with symptoms of decline. In addition, experimental procedures based on culture media are time-consuming and difficult to apply to large sample sizes.

Finally, we would like to emphasize that the sample size required for a proteomics screening to identify candidate biomarkers is different from that required for a widespread application and its validation. The sample size of the present study (six biological replicates, which above or within the range of replicates usually analysed by SWATH-MS, as mentioned above) already reflects some of the biodiversity that characterizes the species, and these results could eventually be extended to hundreds of trees after follow-up studies of validation of the most promising target proteins or biological processes.

The fact that the reviewer mentioned that “the lack of biological replicates can be considered a problem with this experiment” also made us reflect that maybe we were not clear on whether the six replicates quantified by SWATH-MS were technical or biological replicates. In order to clearly communicate that six biological replicates per group were individually quantified by SWATH-MS, we have clarified this number of biological replicates throughout the manuscript text (e.g. methods page 8 and results page 14 where the sentence below was added). 

“The protein profiles were obtained for six individual biological replicates in the two experimental conditions (control and inoculated), to account for some of the genetic diversity between individuals within the Q. suber species.”

[8] Pereira-Leal et al. BMC Genomics 2014, 15:371 http://www.biomedcentral.com/1471-2164/15/371

[9] Hardoim P, Guerra R, Rosa da Costa A, Serrano M, Sánchez M, Coelho A. Temporal metabolic profiling of the Quercus suber–Phytophthora cinnamomi system by middle‐infrared spectroscopy. For Pathol. 2016;46(2):122-33.

[10] Coelho AC, Horta M, Ebadzad G, Cravador A. Quercus suber–P. cinnamomi interaction: hypothetical molecular mechanism model. NZ J Forestry Sci. 2011;(41S):143-57.

[11] Oßwald W, Fleischmann F, Rigling D, Coelho A, Cravador A, Diez J, et al. Strategies of attack and defence in woody plant–Phytophthora interactions. For Pathol. 2014;44(3):169-90.

4 – Why the experiment lasted for 248 days?

Response: In a long-term assay (more than 6 months) the interaction of the plant with the oomycete is settled and the metabolism of the plant would reflect this new phase, different from the initial phase of pathogen recognition and activation of the defence system at the inoculation site. A trial over 6 months gives the plant time and conditions to reorganize its metabolism in Response to an established interaction that will be reflected throughout the plant, and this established differences in the long-term were the focus of this study as reflected in the title of the manuscript. 

Reviewer #2

I have the opportunity to revise your ms on Q. suber-P .cinnamomi interaction. I find the subject highly relevant and of interest for the community. In my view, the analysis of the leaf proteome evaluation of root infection and the long term analysis (> 200 days after inoculation) are particularly interesting.

The proteomic approach is well described, exceptions being the designation of the particular method used and the absence of Q. suber genome database. The meaning of SWATH-MS only appears in lines 233-234 and while authors sustain it is a novel approach (e.g. lines 164-165) its benefits and main features are not explained. Researches in the proteomic field will be able to navigate through this but others will struggle. Authors need to explain why not use the Q. suber genome database.

Response: Dear reviewer, we appreciate the detailed revision of the manuscript and we went through all the questions to reply to the major concerns. In some cases we have grouped the questions in order to answer in a more integrated way.

In order to better explain the principles of the novel SWATH-MS quantitative proteomics approach, its advantages and previous applications in plants, we have added a paragraph summarizing them at the end of the introduction, a new section of Introduction in the reformulated S1 File (now designated “Description of the SWATH-MS, principles and detailed materials and methods”) and six new references.

Besides the analyses of the proteomics data using the three reference proteomes extracted from the curated Uniprot database (Plant, Arabidopsis and Populus), we have also performed both IDA identification and SWATH quantification using as reference the predicted proteins deduced from the draft genome of Quercus suber, downloaded from the CorkOakDB. The results from this analysis are now included in Table 1 and we supply both identification and quantification data in the new Supplementary S3 table.

However, the results of identification and quantification using the cork oak predicted proteins were of low confidence and indicated a high redundancy of the database, making it clear that only the proteomics analyses against a curated reference proteome like that of Arabidopsis, could allow to proceed for functional enrichment analyses and discussion of the biological relevance of differential proteins. This option on the use of the reference proteome is now justified in the results and discussion, below Table 1 in the manuscript.

To better explain the reviewer the redundancy found, we give the example of the protein Heat shock 70-5 (HSP7E/BiP1), accession no. Q9S9N1, for which we detected significantly increased levels in inoculated leaves with 1.6 fold change (FC) increase, Log2FC 0.7 and p value 0.004 using the Arabidopsis reference proteome (Table 4 and S2).

When searching for the equivalent protein in the CorkOak db using Blastp, we found 86 cork oak matches with a significant and stringent Evalue < 10-10:

XP_023873386.1, XP_023913551.1, XP_023895458.1, XP_023905508.1, XP_023919226.1, XP_023905510.1, XP_023899452.1, XP_023911441.1, XP_023911439.1, XP_023907019.1, XP_023909298.1, XP_023883429.1, XP_023907786.1, XP_023897060.1, XP_023885072.1, XP_023907785.1, XP_023916070.1, XP_023909297.1, XP_023895846.1, XP_023911440.1, XP_023923165.1, XP_023913737.1, XP_023919159.1, XP_023919157.1, XP_023909296.1, XP_023891854.1, XP_023883397.1, XP_023901577.1, XP_023914412.1, XP_023892544.1, XP_023928916.1, XP_023920009.1, XP_023896882.1, XP_023895387.1, XP_023895381.1, XP_023901447.1, XP_023897061.1, XP_023870892.1, XP_023873748.1, XP_023879669.1, XP_023918194.1, XP_023913006.1, XP_023902007.1, XP_023925022.1, XP_023886202.1, XP_023899441.1, XP_023925083.1, XP_023880465.1, XP_023918196.1, XP_023883053.1, XP_023896881.1, XP_023909296.1, XP_023913013.1, XP_023926726.1, XP_023925104.1, XP_023918196.1, XP_023883075.1, XP_023900177.1, XP_023896396.1, XP_023885890.1, XP_023907380.1, XP_023912931.1, XP_023903977.1, XP_023925437.1, XP_023871996.1, XP_023872670.1, XP_023892604.1, XP_023905749.1, XP_023905748.1, XP_023929022.1, XP_023909305.1, XP_023886934.1, XP_023905981.1, XP_023870858.1, XP_023897061.1, XP_023917974.1, XP_023880295.1, XP_023904983.1, XP_023882020.1, XP_023891854.1, XP_023907785.1, XP_023907786.1 and XP_023885032.1,.

More than half of these matches (43) had an Evalue of 0, denoting a perfect match of the same Arabidopsis protein with multiple proteins in the cork oak database and confirming the high redundancy found in this database. 

When we did the 2nd confirmation step, searching for the corresponding 43 proteins in the identification (IDA) results for the leaf proteome performed using the cork oak predicted proteins (new supplementary table S3.1), only 8 proteins (18%) could be identified:

N Accession Name

459 XP_023895458.1 heat shock cognate 70 kDa protein 2 [Quercus suber]

459 XP_023899452.1 heat shock cognate 70 kDa protein 2-like [Quercus suber]

69 XP_023873748.1 heat shock 70 kDa protein, mitochondrial [Quercus suber]

725 XP_023905508.1 heat shock cognate 70 kDa protein 2-like [Quercus suber]

1124 XP_023873386.1 heat shock cognate 70 kDa protein 2-like [Quercus suber]

1178 XP_023913737.1 luminal-binding protein 5 [Quercus suber]

524 XP_023923165.1 luminal-binding protein 5-like [Quercus suber]

62 XP_023919226.1 probable mediator of RNA polymerase II transcription subunit 37c [Quercus suber]

Protein 459 matched 2 possible proteins: XP_023899452.1 that was quantified with apparent decreased levels (Log2FC -0.1) but a non-significant p value of 0.39 (S3.2 Table), and the alternative protein XP_023895458.1 matched using the same peptides, that could not be quantified by SWATH (S3.2 Table).

Protein 69 (XP_023873748.1) was quantified with apparent unchanged levels (Log2FC -0.02) and non-significant p value of 0.7 (S3.2 Table).

Protein 725 (XP_023905508.1) was identified with significantly increased levels (Log2FC 0.8 and p value of 0.009) (S3.2 Table).

Protein 1178 (XP_023913737.1) was identified with apparent increased levels (Log2FC 0.5 and non-significant p value of 0.09) (S3.2 Table).

Protein 62 (XP_023919226.1) was identified with significantly decreased levels (Log2FC 0.44 and p value of 0.01) (S3.2 Table).

Finally, proteins 1124 (XP_023873386.1) and 524 (XP_023923165.1) could also not be found in the quantification by SWATH (S3.2 Table).

This detailed analysis was also carried out for all 18 selected differential proteins in which our discussion is focused, and the results obtained confirmed the high redundancy of the cork oak genome database with an average of 26 predicted cork oak proteins matched with Evalue<10-10 for each Uniprot Arabidopsis protein and a maximum of 175. The consequence in terms of SWATH analysis was that most of the predicted cork oak proteins could not be quantified under the quality criteria used for SWATH, as shared peptides are not quantified, or the quantification of a given proteins is spread over several redundant proteins thus giving unreliable results.

However, the comprehensive proteomics work is not supported by a physiological assay, including the success and extension of infection and if mortality was observed. A single photo is presented, in which the rot is mentioned but not very visible.

So, a large part of the biological conclusions, including marker proteins, are unsupported by the presented data. In my view, the publication of the authors’ findings, and their hypothesis about processes trigged by infection requires an extensive revision and the inclusion of such data.

Authors indicate the two groups of plants are not distinguished at the end of the assay. However, the data strongly support s for altered primary metabolism including carbon assimilation. This highlights the need to clearly explain which type of leaf was used for analysis.

Lines 391-395. This part is not clear to me (please consider previous comments). Does homeostatic state mean there is an on-going infection?

Response: The interaction between cork oak and P. cinnamomi does not fit the classic models described for compatible or incompatible interactions. After inoculation of cork oak plants with P. cinnamomi the phenotypic success of colonization is revealed by the appearance of necroses in the site of inoculation (now shown in figure S2 more clearly). In addition, there was already prior knowledge about the effectiveness of the cork oak infection by PA45 isolate, since the histological analysis of cork oak roots colonized by PA45 revealed penetration of the epidermal and subepidermal cell layers and invasion of the cortex, hyphae growing actively within the cortical parenchyma and host cell destruction [1]. 

With amounts of inoculum similar to those used in this assay, plant death is not observed. Plants may eventually die when infesting the soil with large amounts of inoculum and when the plants are submitted to regular flooding to favour root infection [2]. When the lesions on the roots and the disappearance of the fine roots become limiting factors for the development of the plant, symptoms similar to water stress appear in the leaf part. However, it is difficult to observe these symptoms within the time of in vitro assay. 

Similarly, in nature, cork oak trees can show water stress symptoms for many years. The degree of canopy defoliation intensifies over time, the trees lose their vitality, cork harvest is no longer possible, and trees dry up after several years of weakening (cork oak decline). 

The difference observed in the protein profiles of the control plants when compared to the inoculated plants is unequivocal and make evident the effectiveness of the inoculation with P. cinnamomi at the molecular level and over time. 

After inoculation and during infection, the oomycete will be detected by the host that activates the defence system. However, this does not mean that the host is able to limit the progression of the oomycete or the spread of infection to the adjacent roots. Over time, the interaction with the oomycete will promote successive metabolic adjustments in the plant and these changes become evident immediately at the molecular level. 

There is no guarantee that molecular evidences have a corresponding detectable physiological change because under less favourable conditions the plant can use alternative biochemical mechanisms to guarantee its physiological development. For this reason, there would be no guarantee that physiological data would help to justify the observed protein changes that represent the long-term responses.

[1] Horta M, Caetano P, Medeira C, Maia I, Cravador A. Involvement of the β-cinnamomin elicitin in infection and colonisation of cork oak roots by Phytophthora cinnamomi. Eur J Plant Pathol. 2010;127(3):427-36.

[2] Serrano MS, Rios P, Gonzalez M, Sanchez ME. Experimental minimum threshold for Phytophthora cinnamomi root disease expression on Quercus suber. Phytopathol Mediterr. 2015:461-4.

Two acorns per tree were used with a total of six trees. Can authors provide more info about the parentals? Were seeds taken from a natural regeneration stand? How likely it is they came from the same gene pool? In addition, as the parental were showing evidence of decline other questions arise:

a) Which are the causes of the decline?

b) Was the extension of damage similar?

c) How was priming effects addressed in the study?

d) Were only two seed per parental used?

e) Given each parental source, was the kinetics of infection, the amount and severity of symptoms similar?

Response: Table 1 is now a supplementary table and has the GPS references for cork oak acorns progenitors. Parental trees are from the Algarve region and the degree of defoliation presented by the trees varied from 10% to 61%.

In Portugal, most of the genetic variation is comprised within Q. suber populations (96%) while 3.6% is among populations. Differences among populations within geographic regions account for 2.6% of the total variation and only 1.3% is attributed to variation among regions denoting little differentiation of populations over a range of 700 km [3]. It is unlikely that progenitors come from the same gene pool. 

There are no parameters beyond the degree of defoliation to qualify the severity of the decline. Several factors contribute to the cork oak decline, namely, pathogens, soil, genetics, climate change, forest management, but there is no system to assess the relevance of each one of them in this process. In the face of such a complex scenario in the field, it is not easy to reproduce laboratory tests that include all the variables.

[3] Coelho AC, Lima M, Neves D, Cravador A. Genetic diversity of two evergreen oaks [Quercus suber (L.) and Quercus ilex subsp. rotundifolia (Lam.)] in Portugal using AFLP markers. Silvae Genet. 2006;55(1-6):105-18.

At this point, and after analysing Introduction, M&M and results, I recommend extensive revision. Please try to keep your sentences short.

Some detailed comments:

Line 62. Is Portugal the only county affected by cork oak decline?

Response: Cork oak savanna-like ecosystem only exist in Portugal and Spain. So, the reference to Spain was included in the sentence.

Line 69 and following. In the ms, the type of infection (biotrophic, hemibiotrophic or necrotrophic) is absent as well as the typical immune plant Responses it. In my view, it is important to describe the system and will help readers to better follow the discussion.

Response: As it is a proteomics manuscript, we chose to highlight the molecular aspects of the interaction (lines 69-119). The main role of the effector molecules produced by this hemibiotrophic oomycete stands out, as well as the defence models known in chestnut and cork oak.

Line 86. RxLr domain of?

Response: The paragraph is about the mechanism of entry of effector molecules that have the RxLR domain. It is inferred that it will be the domain of these molecules.

Lines 87-90. Reprogramming in what way? Are HR and PCD promoted upon infection? If I recall properly, higher PDC and HR are typical of plant resistance Responses.

Response: Effector molecules secreted by the oomycete are recognized by Q. suber receptors triggering a hypersensitive Response and PCD. Activation of the defence system it is not synonymous of oomycete resistance or capability to prevent the progression of the pathogen. For more information, please see references 4 for cork oak and 5, 6 for Castanea sativa. 

4. Oßwald W, Fleischmann F, Rigling D, Coelho A, Cravador A, Diez J, et al. Strategies of attack and defence in woody plant–Phytophthora interactions. For Pathol. 2014;44(3):169-90.

5. Serrazina S, Santos C, Machado H, Pesquita C, Vicentini R, Pais MS, et al. Castanea root transcriptome in Response to Phytophthora cinnamomi challenge. Tree Genet Genomes. 2015;11(1):6.

6. Santos C, Duarte S, Tedesco S, Fevereiro P, Costa RL. Expression profiling of 

Castanea genes during resistant and susceptible interactions with the oomycete pathogen Phytophthora cinnamomi reveal possible mechanisms of immunity. Front Plant Sci. 2017;8:515.

Line 91. Please explain briefly compatible and incompatible interactions. Please add more info about the known mechanisms in your system. What happens in your dataset? If info is inexistent or contradictory, please add such info.

Lines 91-100. Please make this section more clear. In its current version, PTI and ETI strategies are mixed.

Lines 133-136. Relevance?

Response: the hypothetical molecular models referred to in lines 133 and 136 have the answer to the addressed questions. In the interactions between Q. suber or C. sativa and P. cinnamomi we cannot classify the interaction only as compatible or as incompatible and, therefore, we cannot adapt the text to these conventions. Perhaps it would be suitable for other plant species but not for these trees and for interactions that do not follow the standards.

Line 155. As authors make leaf extracts, how is this approach non-invasive?

Response: Leaf extracts were made at the end of the assay. The objective is to expand this screening to trees in the forest systems that have thousands of leaves. The sentence has been rewritten: The leaves are a distal organ that can be sampled in a minimally invasive way in adult trees….

Lines 164-165. Relevance?

Response: They were removed

Lines 173. Please described which type of leaf was used and if they were asymptomatic. Add the methodology used to quantify it or score it. It would be important to have leaf characterization as leaf age, area or biomass. 

Lines 305-307. Please add the quantitative data or scoring matrix.

Response: The plants did not show chlorosis of the leaves, brown spots or other disease symptoms.

The extraction of proteins was made from a mixed pool of full-expanded leaves (200g).

Line 181. Please provide GPS coordinates.

Response: These have been included in Table S1.

Line 183. Distinct developments stages. Such as?

Response: Replaced by " at distinct stages of progression".

Line 185. Please provide strain characteristics.

Response: P. cinnamomi, isolate PA45, mating type A2. The isolate is referenced in several publications [1]. 

[1] Horta M, Caetano P, Medeira C, Maia I, Cravador A. Involvement of the β-cinnamomin elicitin in infection and colonisation of cork oak roots by Phytophthora cinnamomi. Eur J Plant Pathol. 2010;127(3):427-36.

Lines 197-199. How was the success of infection established? Please provide details on the soil characteristics and soil nutrition as mineral availability has an impact on the oomycete.

Response: Agar plugs of P. cinnamomi mycelium isolate PA45, grown in clarified V8 (Campbell Soup) semi-solid agar, in the dark at 25 ºC [11] for 9 days, was placed mycelial surface down on the tap root of 6 cork oak plants.

The oomycete contacted directly with root tissues. 

Line 237, line 257 & S1 file. Authors should run the data on a more recent database. The dbases were used more than two years before submission.

Response: We acknowledge the observation that whenever possible the results should be analysed with the more updated versions of the databases, available at the time that the study is carried out. 

Indeed, we have obtained the proteomics results in 2018 and used for the enrichment analyses the updated gene ontologies and pathways downloaded in November 2017. 

It would not make sense to run all the analyses by the time the article is submitted, as all the analyses and discussion take time and the obtained results are clearly indicated in the methods with which DB and accession date were used, in case some researchers intend to repeat; Nevertheless, in order to confirm if similar results would be obtained if the study started today we have repeated the enrichment analyses using the Arabidopsis gene ontology and pathways downloaded in November 2020. As we can see in the table below, although the precise designation given to each group of enriched terms may be slightly changed, the same enriched processes are in general obtained and related to the changes in the same proteins. 

Since the discussion is guided by the identity of particular proteins whose levels were indeed changed with the challenge, rather than on the prediction of processes/pathways enriched, we do not see the need to alter the GO/pathway enrichment results provided in supplementary tables, as these would cause an unnecessary reorganization of the discussion. Nevertheless, we indicate in the methods that "enrichment analyses were repeated using the databases updated in 2020 and the same general enriched terms were found (data not shown)".

File S1. Please explain why to exclude peptides with biological modifications. What is a biological modification, a PTM?

Response: A biological modification is in fact a PTM, and peptides identified with these modifications are not used in an untargeted analysis of the total level of expression of the proteins, once these modified peptides may lead to distinct expression patterns that are only indicative of the expression of the modification and not of the total level of expression of the protein.

Line 241. Per plant, eight replicate MS-runs, i.e. 2 groups x 6 plants x 8 replicate runs?

Response: It has been corrected. 2 groups × 6 plants.

Six biological replicates × 2 groups.

Line 307-309. How authors deal with this issue and how it impacts the findings of this ms?

Response: The criteria adopted for the selection of the set of differential proteins was sufficiently strict, avoiding distortions originating from factors such as genetic diversity. 

Line 326-327. How does this variability relate to the ones found in other studies? In my view, the sentence is too simplistic and needs revision. Several factors can support such differences.

Response: In similar studies, genetic diversity is avoided through the use of half-siblings plants.

Lines 293-297. The info needs to be presented much earlier in the ms.

Response: The following information now appears in the material and methods.

"PA45 was isolated from the rhizosphere of cork oak trees that showed symptoms of decline in the Algarve region and its high virulence on cork oak seedlings was extensively studied"

Line 328-329. Something seems to be missing in the sentence. Please confirm.

Response: The sentence has been rewritten.

Line 280-1. Was infection at this point confirmed? Biomass and growth parameters are highly relevant for all the remaining discussion.

Lines 396-398. This part is not clear to me.

Lines 398-400. Idem.

Response: The evaluation of root infection by P. cinnamomi is usually done in a qualitative way and is not very reliable because it depends on the observer. This information is now included in the document. In addition, in vitro procedures have no applicability in the field on adult trees. Assessing cork oak decline in the field is based on the degree of canopy defoliation, and even if P. cinnamomi is isolated from the roots of declining trees, it is not possible to know the level or time of infection. Observing the roots to detect the presence of necrosis promoted by the oomycete or isolating the pathogen at the end of the in vitro assay would only reinforce the message that the plants were infected and would not add information about the defence responses triggered by each of the 6 biological replicates. What was important was to guarantee that P. cinnamomi had interacted with the cork oak and this was ensured through the chosen and validated inoculation procedures [1]. The presence of necrosis at the inoculation site confirm that the oomycete had invaded the host tissues and that it had triggered a Response reaction. It is important to have a group of molecular markers that are indicative of a potential infection by P. cinnamomi that does not depend on the identification or quantification of the oomycete in the rhizosphere of an adult tree.

[1] Horta M, Caetano P, Medeira C, Maia I, Cravador A. Involvement of the β-cinnamomin elicitin in infection and colonisation of cork oak roots by Phytophthora cinnamomi. Eur J Plant Pathol. 2010;127(3):427-36.

Lines 441. Highest and lowest scores mean?

Response: In the Materials and Methods, section of Enrichment analyses and hierarchical clustering, we have defined “Enrichment scores of the functionally related network groups were calculated as -Log2 [group FDR]”. In order to make it more clear, this sentence was rewritten as “Comparing the results from KEGG and REACTOME pathways, the protein subset is enriched in Ribosome and SRP-dependent cotranslational protein targeting to membrane, with the highest enrichment scores (lowest FDR), respectively, and Glycolysis/Gluconeogenesis and Glucose metabolism with the lowest enrichment scores, respectively”.

Lines 464-465. The sentence is not clear to me.

Response: The variation pattern of the proteins associated to a GO-BP group can be the same in all (for example: more abundant in the inoculated plants) or be variable, that is, 1 protein of that group is more abundant in the inoculated samples and another protein of the same group is less abundant in the inoculated samples.

Lines 493-494. Your observation made me wonder about the values of the protein for the same parental origin. Could parental origin be a factor for the difference?

Response: No, because we have considered the median of the 6 samples.

Lines 503. RBCS1S and RBCS3C are both from At. Please confirm. Also, please check for consistency in RubisCO designation (line 510, 512, elsewhere?).

Response: RBCS1A and RBCS3B are two major members within the Arabidopsis RBCS multigene family. RubisCO designation is corrected in the new document.

Lines 508-509. A possible explanation for it?

Response: Activation of alternative pathways.

Lines 510-520. In my view, this paragraph is not in line with the previous one.

Response: There seems to be an agreement in the results presented in the 2 paragraphs - reduced photosynthetic activity - decrease in the accumulation of carbon assimilated.

Lines 523-527. The inclusion of this sentence at this point is not clear to me.

Throughout the discussion. Data points out for lower assimilation capacity but authors also indicate plants do not differ in growth and appearance.

Response: The results obtained were presented and an attempt was made to interpret biologically.

Line 580. Energy production needed for?

Response: Included " associated with the immune response."

Line 734-735. The sentence is not clear to me.

Response: We are only presenting the results obtained.

Line 736. Does long term defence imply that plants cope successfully with the infection or not?

Response: What we see in adult plants is a slow decline in which the plants lose their vitality over many years (10 or more). Therefore, it appears that plants do not successfully deal with infection.

Lines 741-742. In my view, authors need to better support the assay with physiological observations as well as cytological (see initial comments).

Line 745. I do not agree that the immune Response was observed. The authors analyse the proteome.

Lines 746-747. See previous comments.

Lines 754-777. In my view, the statements are not adequate in face of the present data.

Response: The answers to these questions have been addressed previously.

Figure 1. Is more suitable as graphical Abstract. Type of leaf should be added as well as the main results. As figure is not very informative.

Response: Figure 1 was removed according to the suggestion of other reviewers.

Figure 2. Symptoms not visible. The scale should be added.

Response: The quality of figure 2 was improved. A 2 cm2 agar plug of P. cinnamomi mycelium was used (information added to material and methods). The size of the lesion is about 2 cm long. Figure 2 is now in the supplementary material such as S2 Fig.

Figure 3. As figure is not very informative. Add it to a graphical Abstract?

Response: It is considered a good suggestion but it was not adopted in this case.

Reviewer #3

I recommend and insist on using the Cork oak database instead of Arabidopsis data base to identify proteins

Otherwise, it I cannot proceed with the rest of paper until we receive the new list of identified protein using the Q. suber data base and of course the related information.

Response: Dear reviewer, we appreciate the detailed revision of the manuscript and we went through all the questions to reply to the major concerns. 

Besides the analyses of the proteomics data using the three reference proteomes extracted from the curated Uniprot database (Plant, Arabidopsis and Populus), we have also performed both IDA identification and SWATH quantification using as reference the predicted proteins deduced from the draft genome of Quercus suber, downloaded from the CorkOakDB. The results from this analysis are now included in Table 1 and we supply both identification and quantification data in the new Supplementary S3 table.

However, the results of identification and quantification using the cork oak predicted proteins were of low confidence and indicated a high redundancy of the database, making it clear that only the proteomics analyses against a curated reference proteome like that of Arabidopsis, could allow to proceed for functional enrichment analyses and discussion of the biological relevance of differential proteins. This option on the use of the reference proteome is now justified in the results and discussion, below Table 1 in the manuscript.

To better explain the reviewer the redundancy found, we give the example of the protein Heat shock 70-5 (HSP7E/BiP1), accession no. Q9S9N1, for which we detected significantly increased levels in inoculated leaves with 1.6 fold change (FC) increase, Log2FC 0.7 and p value 0.004 using the Arabidopsis reference proteome (Table 4 and S2).

When searching for the equivalent protein in the CorkOak db using Blastp, we found 86 cork oak matches with a significant and stringent Evalue < 10-10:

XP_023873386.1, XP_023913551.1, XP_023895458.1, XP_023905508.1, XP_023919226.1, XP_023905510.1, XP_023899452.1, XP_023911441.1, XP_023911439.1, XP_023907019.1, XP_023909298.1, XP_023883429.1, XP_023907786.1, XP_023897060.1, XP_023885072.1, XP_023907785.1, XP_023916070.1, XP_023909297.1, XP_023895846.1, XP_023911440.1, XP_023923165.1, XP_023913737.1, XP_023919159.1, XP_023919157.1, XP_023909296.1, XP_023891854.1, XP_023883397.1, XP_023901577.1, XP_023914412.1, XP_023892544.1, XP_023928916.1, XP_023920009.1, XP_023896882.1, XP_023895387.1, XP_023895381.1, XP_023901447.1, XP_023897061.1, XP_023870892.1, XP_023873748.1, XP_023879669.1, XP_023918194.1, XP_023913006.1, XP_023902007.1, XP_023925022.1, XP_023886202.1, XP_023899441.1, XP_023925083.1, XP_023880465.1, XP_023918196.1, XP_023883053.1, XP_023896881.1, XP_023909296.1, XP_023913013.1, XP_023926726.1, XP_023925104.1, XP_023918196.1, XP_023883075.1, XP_023900177.1, XP_023896396.1, XP_023885890.1, XP_023907380.1, XP_023912931.1, XP_023903977.1, XP_023925437.1, XP_023871996.1, XP_023872670.1, XP_023892604.1, XP_023905749.1, XP_023905748.1, XP_023929022.1, XP_023909305.1, XP_023886934.1, XP_023905981.1, XP_023870858.1, XP_023897061.1, XP_023917974.1, XP_023880295.1, XP_023904983.1, XP_023882020.1, XP_023891854.1, XP_023907785.1, XP_023907786.1 and XP_023885032.1,.

More than half of these matches (43) had an Evalue of 0, denoting a perfect match of the same Arabidopsis protein with multiple proteins in the cork oak database and confirming the high redundancy found in this database. 

When we did the 2nd confirmation step, searching for the corresponding 43 proteins in the identification (IDA) results for the leaf proteome performed using the cork oak predicted proteins (new supplementary table S3.1), only 8 proteins (18%) could be identified:

N Accession Name

459 XP_023895458.1 heat shock cognate 70 kDa protein 2 [Quercus suber]

459 XP_023899452.1 heat shock cognate 70 kDa protein 2-like [Quercus suber]

69 XP_023873748.1 heat shock 70 kDa protein, mitochondrial [Quercus suber]

725 XP_023905508.1 heat shock cognate 70 kDa protein 2-like [Quercus suber]

1124 XP_023873386.1 heat shock cognate 70 kDa protein 2-like [Quercus suber]

1178 XP_023913737.1 luminal-binding protein 5 [Quercus suber]

524 XP_023923165.1 luminal-binding protein 5-like [Quercus suber]

62 XP_023919226.1 probable mediator of RNA polymerase II transcription subunit 37c [Quercus suber]

Protein 459 matched 2 possible proteins: XP_023899452.1 that was quantified with apparent decreased levels (Log2FC -0.1) but a non-significant p value of 0.39 (S3.2 Table), and the alternative protein XP_023895458.1 matched using the same peptides, that could not be quantified by SWATH (S3.2 Table).

Protein 69 (XP_023873748.1) was quantified with apparent unchanged levels (Log2FC -0.02) and non-significant p value of 0.7 (S3.2 Table).

Protein 725 (XP_023905508.1) was identified with significantly increased levels (Log2FC 0.8 and p value of 0.009) (S3.2 Table).

Protein 1178 (XP_023913737.1) was identified with apparent increased levels (Log2FC 0.5 and non-significant p value of 0.09) (S3.2 Table).

Protein 62 (XP_023919226.1) was identified with significantly decreased levels (Log2FC 0.44 and p value of 0.01) (S3.2 Table).

Finally, proteins 1124 (XP_023873386.1) and 524 (XP_023923165.1) could also not be found in the quantification by SWATH (S3.2 Table).

This detailed analysis was also carried out for all 18 selected differential proteins in which our discussion is focused, and the results obtained confirmed the high redundancy of the cork oak genome database with an average of 26 predicted cork oak proteins matched with Evalue<10-10 for each Uniprot Arabidopsis protein and a maximum of 175. The consequence in terms of SWATH analysis was that most of the predicted cork oak proteins could not be quantified under the quality criteria used for SWATH, as shared peptides are not quantified, or the quantification of a given proteins is spread over several redundant proteins thus giving unreliable results.

Besides, we have some comments concerning the

Introduction: you have written a very long introduction mentioning a several results of others researchers, it seem a part of a review, please reduce it and mention only the related information with your paper and adding a section describing the SWATH-MS quantitative proteomics used, advantage and relative works

Response: In order to better explain the principles of the novel SWATH-MS quantitative proteomics approach, its advantages and previous applications in plants, we have added a paragraph summarizing them at the end of the introduction, a new section of Introduction in the reformulated S1 File (now designated “Description of the SWATH-MS, principles and detailed materials and methods”) and six new references. Information was removed from the introduction in order to reduce the text size. 

Materials and methods: we don´t understand the importance of the mentioned information about the seeds and table 1, please rewrite this section clearly and provide table 1 as supplementary materials

Response: Table 1 is now a supplementary table S1 Table. The Biological material section of MM has been rewritten and includes additional information.

Results and discussion

In the first part of this section, you are describing the effect of inoculation after 24h and 48 h, then after 7 months. You have demonstrating the roots on the two first point of time (fig. 1 a, b) however, you didn’t do with 7months. Please rewrite this part because it is very confused, and it is difficult to understand the meaning of the sentences.

Response: After inoculation of cork oak plants with P. cinnamomi the phenotypic success of colonization is revealed by the appearance of necroses in the site of inoculation (now shown in figure S2 more clearly). In addition, there was already prior knowledge about the effectiveness of the cork oak infection by PA45 isolate, since the histological analysis of cork oak roots colonized by PA45 revealed penetration of the epidermal and subepidermal cell layers and invasion of the cortex, hyphae growing actively within the cortical parenchyma and host cell destruction [1]. 

With amounts of inoculum similar to those used in this assay, plant death is not observed. Plants may eventually die when infesting the soil with large amounts of inoculum and when the plants are submitted to regular flooding to favour root infection [2]. When the lesions on the roots and the disappearance of the fine roots become limiting factors for the development of the plant, symptoms similar to water stress appear in the leaf part. 

Usually, the evaluation of root infection by P. cinnamomi is done in a qualitative way and is not very reliable because it depends on the observer. This information is now included in the document. The difference observed in the protein profiles of the control plants when compared to the inoculated plants at the end of the assay is unequivocal and make evident the effectiveness of the inoculation with P. cinnamomi at the molecular level and over time. 

Observing the roots to detect the presence of necrosis promoted by the oomycete or isolating the pathogen at the end of the in vitro assay would only reinforce the message that the plants were infected and would not add information about the defence Responses triggered by each of the 6 biological replicates. Furthermore, it is important to have a group of molecular markers that are indicative of a potential infection by P. cinnamomi that does not depend on the identification or quantification of the oomycete in the rhizosphere of an adult tree. Over time, the interaction with the oomycete will promote successive metabolic adjustments in the plant and these changes become evident immediately at the molecular level. 

[1] Horta M, Caetano P, Medeira C, Maia I, Cravador A. Involvement of the β-cinnamomin elicitin in infection and colonisation of cork oak roots by Phytophthora cinnamomi. Eur J Plant Pathol. 2010;127(3):427-36.

[2] Serrano MS, Rios P, Gonzalez M, Sanchez ME. Experimental minimum threshold for Phytophthora cinnamomi root disease expression on Quercus suber. Phytopathol Mediterr. 2015:461-4.

Figures, a very poor quality of figures are provided and lack of information like the figure 3

Response: The quality of the figures was improved.

What is the importance of figure 1? eliminate it

Response: We have removed this figure as suggested.

Reviewer #4

We thank the reviewer for their clear and positive assessment of our MS and for taking the time to read and provide constructive comments for improvement.

Maybe title as 

Disclosing proteins in cork oak leaves associated with the immune Response of cork oak inoculated in the roots with P c: a long-term assay 

Response: We followed the suggestion and the manuscript is now entitled: “Disclosing proteins in the leaves of cork oak plants associated with the immune response to Phytophthora cinnamomi inoculation in the roots: a long-term proteomics approach”

I do not think the Figure is necessary, as the necessary information is captured clearly by the sentences which follow? Consider removing this Figure

Response: Figure 1 was removed.

How far apart were the 6 oak trees from each other. GPS references, or reference?

Response: GPS references are shown in table S1 of the supplementary documentation.

Maybe give the Genbank number, so you can confirm definitely P. c

Response: Information included in the Biological Material section: To reconfirm the identity of the isolate as P. cinnamomi, DNA was extracted from PA 45 isolate and was used in PCR reactions with primers (95.422/96.007) designed for a colorimetric molecular assay [1] targeting the elicitin genes (GenBank accession number AJ000071).

[1] Coelho, AC, Cravador, A, Bollen, A, Ferraz, JFP, Moreira, AC, Fauconnier, A, et al. Highly specific and sensitive non-radioactive molecular identification of Phytophthora cinnamomi. Mycol. Res. 1997; 101(12):1499-1507

What type of agar (PDA, V8 or ? and how old was the mycelium? 

Maybe rewrite this bit as ‘----- isolate PA45 grown on XXX agar for 7 days at 25C was placed mycelial surface face down on the tap root of 77-day-old cork oak plants’. ???

Importantly, how were the plants maintained for that 48 hours, kept in a humid chamber, misted, or what? Information is needed, so others can repeat what you did.

Were the inoculum plugs removed at this time or not? Please include this information

Substrate seedlings were placed into?

We also need detail of how you ‘sham’ inoculated the control plants; did you just use non-colonised agar? Please provide the detail.

Delete this sentence as said two lines down.

Response: All the information requested was included in the Biological material section.

You imply that this is a single inoculation, here and elsewhere in the text. However, it is very likely the pathogen under your watering regime continued to produce sporangia and release zoospores that would have continued to infect new roots as they were produced. Or alternatively, root to root contact could have resulted in infection from necrotic roots to healthy roots. This scenario needs to be considered. 

Response: The following paragraph has been included in the "Observation of the plants" section: The vegetative development of the plants was similar in both experimental conditions, inoculated and non-inoculated. Although no foliar symptoms of P. cinnamomi infection are observed, the infection is expected to have spread beyond the inoculation site through zoospores released from sporangia who migrated into the irrigation water or through root to root contact.

Do you mean ‘six’?

Response: Yes. The value has been corrected.

This implies you removed the mycelium (colonized agar plug) after 48 hours? If this is the case (or not) this detail needs to be included in the Methods. 

Response: The information is included in the MM - Biological material

So the control plants did not wilt, I would expect they might given you have removed them from the container substrate and inoculated them?

Response: For the preparation of control and inoculated plants, twelve 77-day-hold cork oak plants were removed from the germination alveoli, freeing most of the organic substrate that accompanied the roots, and were laid down on trays whose surface was protected with moist absorbent paper. After 48 h, the control plants did not show the aerial apex wilted.

This image showing the lesion is not clear, as very difficult to see the necrotic tissue. It would also be useful to show example of control plants and a healthy root as comparison. 

Response: The quality of the photos has been improved. As we did not observe any change in the roots of control plants, we did not record them photographically.

It would have been very valuable at the end of the experiment to have washed the container substrate off the roots and compared the inoculated roots with the non-inoculated roots. It is very likely that the tops could look unchanged but you would have had reduced biomass (fine roots and coarse roots) – you could have even done dry weights on the roots. I have experienced/observed this to be the case on a number of different host species – and put it down to the growing conditions, continued watering of plants allows them to maintain a healthy appearance ‘on top’ but actually quite diseased/symptomatic below ground. This would have been quite informative, I think1

Response: We agree with the observation made, however, in previous tests it was never possible to correlate statistically the dry weight of the roots or leaves with the infection, except when infestations were made with large amounts of inoculum. The effectiveness of the infection was always determined by observing the necrotic tissue in the roots.

Agree, could be very valuable indeed. I think where your research is going is exciting and pertinent for Phytophthora researchers. It would be interesting here, or elsewhere in the text to discuss logistics of this approach with respect to expertise required and also the cost of doing the work. So the reader gets an idea of how soon such methods might be available. 

Response: The following paragraph has been added to R&D - Hierarchical clustering of differentially produced cork oak proteins

Furthermore, the current methods used to isolate and identify P. cinnamomi from the rizosphere of oak roots are based on baiting tecnhiques, pathogen growth in selective media and molecular identification with specific primers. These procedures are time consuming, require expertise and are of relatively low effectiveness.

Yes, absolutely in any future work. As would be looking at the roots and their health be important to do.

Response: We agree that it will be important in a future work to include the evaluation of the roots.

A long sentence and quite hard to follow?

Response: It was rewritten.

---

## [Editor Report · Decision Letter 1]

23 Dec 2020

Disclosing proteins in the leaves of cork oak plants associated with the immune response to Phytophthora cinnamomi inoculation in the roots: a long-term proteomics approach

PONE-D-20-29356R1

Dear Dr. Coelho,

We’re pleased to inform you that your manuscript has been judged scientifically suitable for publication and will be formally accepted for publication once it meets all outstanding technical requirements.

Kind regards,

Sara Amancio

Academic Editor

PLOS ONE
---

## [Editor Report · Acceptance letter]

4 Jan 2021

PONE-D-20-29356R1 

Disclosing proteins in the leaves of cork oak plants associated with the immune response to *Phytophthora cinnamomi* inoculation in the roots: a long-term proteomics approach 

Dear Dr. Coelho:

I'm pleased to inform you that your manuscript has been deemed suitable for publication in PLOS ONE. Congratulations! Your manuscript is now with our production department. 

Kind regards, 

on behalf of

Prof Sara Amancio 

Academic Editor

PLOS ONE